# HeLutNet: Extremely Fast Privacy-preserving Inference in Milliseconds via LUT-based Machine Learning Models

## Abstract

Fully Homomorphic Encryption (FHE) holds immense promise for enabling privacy-preserving machine learning (PPML) inference, allowing computations on encrypted data without needing to expose sensitive information to untrusted third parties, such as cloud providers. Despite this potential, a major obstacle to FHE's widespread adoption is its prohibitively high computational overhead and slow inference speeds. For instance, the current state-of-the-art implementation requires 0.23 seconds for a single MNIST inference on one CPU core, a significant slowdown that renders FHE impractical for real-world applications. To overcome this, we introduce HeLutNet, an ultra-low-latency framework that accelerates FHE-based PPML inference by pioneering the use of lookup table (LUT)-based ML models. LUT-based models are uniquely suited for FHE, as they can represent complex non-linear functions without deep network architectures, enabling high accuracy on diverse datasets with shallow networks. Our approach converts a PyTorch-trained, LUT-based model into an efficient FHE program, implementing LUTs using the BGV scheme's SIMD capabilities and incorporating novel packing and LUT reordering strategies to minimize the number of homomorphic operations performed. Our evaluation across 20 vision, speech, health, and tabular datasets demonstrates consistent speedup and lower memory and communication overhead compared to state-of-the-art FHE-based PPML methods. Notably, we reduce inference time on various datasets to just 8.3 ms on a single-core CPU, achieving a 27.71x speedup over Orion (ASPLOS'25), thereby demonstrating the potential for a practical, real-time FHE-based PPML system.

## 1 Introduction

Machine learning (ML) has enabled the development of numerous applications across various domains, including education, healthcare, and finance. In many of these fields, the data is highly confidential. For example, a model might be used to detect a disease from X-ray images, process biometric data, or classify emails. Given the growing concerns over data privacy and the rise of regulations like the General Data Protection Regulation (GDPR) in Europe and the California Consumer Privacy Act (CCPA) in the U.S., there is a pressing need for methods that can protect user data during ML inference.

To address this challenge, researchers have proposed privacy-preserving ML (PPML) inference techniques based on Fully Homomorphic Encryption (FHE). FHE allows computations to be performed directly on encrypted data, which eliminates the risk of data leaks. However, the security provided by FHE comes with a significant computational overhead and high latency. For instance, a single inference on an MNIST image can take 0.23 seconds, while a single CIFAR-10 image takes 3-10 minutes on a single-core Intel Xeon CPU (Ebel et al. (2025)). This high computational complexity and latency have hindered the widespread adoption of FHE-based PPML in real-world applications.

Performing encrypted ML inference using FHE presents numerous challenges. First, FHE-based arithmetic operations are orders of magnitude slower than their unencrypted counterparts, as demonstrated in Tab. 1. Second, most existing FHE schemes support only a limited set of operations, primarily simple linear ones like addition and multiplication. Consequently, nonlinear operations, such

Table 1: Execution time of some FHE operations based on Microsoft SEAL (SEAL) and Lattigo (lat (2024)) implementation of the BFV FHE scheme (ring degree=$2^{14}$). All results are taken from Mouchet et al. (2020).

| Operation | Runtime (ms) | |
|---|---|---|
| | **SEAL** | **Lattigo** |
| Addition | 0.4 | 0.2 |
| Mult w/plain | 8.8 | 7.4 |
| Mult | 65.7 | 44.9 |
| Square | 48.4 | 35.0 |

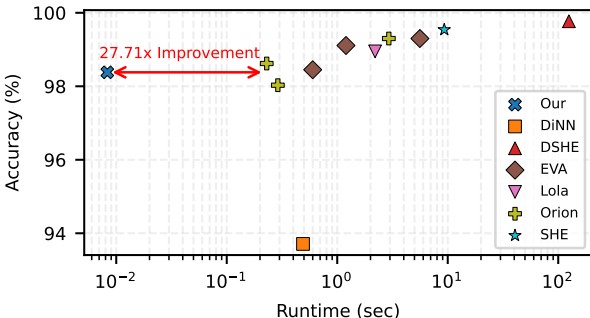

Figure 1: Accuracy and runtime of HeLutNet vs existing FHE-based privacy preserving ML inference solutions on MNIST dataset

as the activation functions in neural networks, must be approximated using polynomial functions (Dowlin et al. (2016); Brutzkus et al. (2019); Chou et al. (2018)) or network architectures must be redesigned to minimize the use of nonlinear activations (Ghodsi et al. (2020); Jha et al. (2021)), often leading to a degradation in accuracy compared to the original models. Third, performing operations on encrypted data (ciphertext) increases the noise level. If a computation is too deep, the noise can grow so high that the ciphertext can no longer be decrypted correctly. To support indefinitely deep calculations, a bootstrap operation (Gentry (2010)) can be performed to reset the noise level. However, bootstrapping is extremely time-consuming, taking from seconds to several minutes depending on the FHE scheme and the required precision (Badawi & Polyakov (2023)). For this reason, many studies have focused on reducing the number and optimizing the placement of bootstrap operations (Ebel et al. (2025); Krastev et al. (2024)), as well as reducing the complexity and runtime of the bootstrap operation itself (Kim et al. (2024); Cheon et al. (2025); Ma et al. (2024)).

Over the years, numerous studies have aimed to improve the accuracy of FHE-based ML inference to match unencrypted versions, reduce runtime overhead, and scale FHE to larger datasets like ImageNet (Ebel et al. (2025); Roy & Roy (2025); Krastev et al. (2024); Dathathri et al. (2020); Lou & Jiang (2019)). Nevertheless, the latency of state-of-the-art fully homomorphic ML inference systems remains in the range of seconds to minutes on small datasets like MNIST, as shown in Figure 1. This latency is an order of magnitude higher than that of ordinary ML inference, making real-time privacy-preserving ML still impractical for widespread deployment.

This paper introduces HeLutNet, a novel framework for ultra-low-latency FHE-based ML inference. For the first time, we propose lookup table (LUT)-based ML models as an alternative, FHE-friendly approach to accelerate PPML inference. LUT-based models are well-suited for FHE because they can represent complex, nonlinear functions without requiring deep architectures or activation functions. This enables high accuracy on diverse datasets using shallow networks (Bacellar et al. (2024); Petersen et al. (2022a)). This key property allows us to bypass the need for approximate activation functions, which caused accuracy degradation in prior works, while also eliminating the computationally intensive and time-consuming bootstrap operations required by other deep architectures.

HeLutNet is implemented as a Python library that converts a trained PyTorch LUT-based ML model into an efficient FHE program, utilizing the Microsoft SEAL (SEAL) or Lattigo (lat (2024)) library as the backend. We derived a novel method to efficiently implement LUTs using integer arithmetic based on the BGV FHE schemes (Brakerski et al. (2014)). Our implementation strategy minimizes the use of expensive rotation operations and maximizes the utilization of the ciphertext encoding space to perform all encrypted computations in a Single Instruction, Multiple Data (SIMD) manner.

We make the following contribution in this work:

1. We have developed and open-sourced HeLutNet, a framework for ultra-low-latency FHE-based ML inference. We are the first to propose LUT-based ML models as a more FHE-friendly algorithm to accelerate privacy-preserving ML inference. We believe this work will open a new line of research focused on optimizing the runtime and accuracy of weightless/LUT-based ML models for privacy-preserving applications.

2. We derive a method to efficiently implement LUT-based ML models within the leveled BGV FHE schemes. Our approach fully utilizes the ciphertext encoding space to leverage BGV's SIMD encrypted arithmetic computation and minimizes the use of expensive rotation operations.

3. We significantly reduce inference time across various real-world vision, speech, healthcare, and tabular datasets to as low as 8.3 ms using a single-core CPU. On MNIST dataset, this represents a 27.71x improvement over the current state-of-the-art, paving the way for real-time fully homomorphic inference on commodity hardware without the need for custom hardware acceleration.

## 2 BACKGROUND AND RELATED WORK

### 2.1 PRIVACY-PRESERVING MACHINE LEARNING

Privacy-preserving machine learning (PPML) enables model training/inference on private data, addressing security and privacy concerns in machine learning. Secure Multi-party computation (MPC) is one promising technique which allowing a set of mutually distrusting parties to jointly compute a function. Many PPML system has been proposed based on the MPC concept with varying security guarantee. Early work explores semi-honest PPML protocol (Mohassel & Zhang (2017); Knott et al. (2021); Koti et al. (2021b)) guarantee secure when all parties honestly follow the protocol but tries to learn secret values (by analyze messages they receive). Maliciously honest majority PPML (Wagh et al. (2021); Koti et al. (2021a)) provides stronger security guarantee against some party perform arbitrarily malicious action during the execution (e.g., sending incorrect values). Note that in both scenario, we assume there are more honest parties than malicious parties which might not satisfy security requirement in realworld scenario. With that, maliciously dishonest majority PPML has been proposed by Damgård et al. (2019) to provide security guarantee for a case when more than half of the party are malicious. While this approach greatly enhance the security, its runtime is 1–3 orders of magnitude slower than prior MPC approach Dalskov et al. (2020). While the overhead of this approach has been recently improved significantly by Yuan et al. (2024), MPC-based in general still has fundamental limitation including the need to have multiple involve party, the high communication cost, and the possibility of leaking some partial information to mallicious party.

### 2.2 FULLY HOMOMORPHIC ENCRYPTION (FHE)

FHE is a powerful encryption scheme that allows for computation to be performed directly on encrypted data. This makes FHE particularly well-suited for implementing PPML in scenarios with highly sensitive data and when there is severely limited trust between parties.

#### 2.2.1 BFV/BGV SCHEME

The Brakerski/Fan-Vercauteren (BFV) scheme (Fan & Vercauteren (2012); Brakerski & Vaikuntanathan (2014)) is a widely used fully homomorphic encryption (FHE) scheme due to its balance between security and performance. The BFV scheme operates on a polynomial ring defined as

$$R = \mathbb{Z}_p[x]/(x^n + 1),$$

where $\mathbb{Z}_p = \mathbb{Z}/p\mathbb{Z}$ and $n$ is a power of two. If $p$ and $n$ are chosen so that a $2n$-th root of unity exists in $\mathbb{Z}_p$, each element of $R$ can be viewed as a vector of length $n$ over $\mathbb{Z}_p$. This vector representation allows the encrypted data (ciphertexts) to support *element-wise operations*, meaning that computations such as addition and multiplication are applied to each corresponding element of two ciphertext vectors independently.

Homomorphic addition and multiplication in BFV directly mirror their plaintext counterparts. Ciphertext addition corresponds to element-wise addition of the underlying plaintext vectors, while ciphertext multiplication corresponds to element-wise multiplication. These fundamental operations form the basis for more complex encrypted computations while preserving data confidentiality.

Another key feature of BFV is the *rotation* operation, which cyclically shifts the positions of elements in an encrypted vector. Rotations can move data by a chosen number of positions, enabling more advanced operations such as matrix multiplications and convolutions to be performed directly on ciphertexts.

## 2.3 LUT-BASED MACHINE LEARNING MODEL

A LUT-based ML model, a.k.a. weightless neural network (WNN), is a multiplication-free network that uses binary lookup tables (LUTs) instead of weighted connections. Early work focused on converting neurons to LUTs (Umuroglu et al. (2020)), later improving efficiency by packing multiple $n$ neurons into one LUT (Andronic & Constantinides (2023; 2024)). However, a fundamental limitation of this conversion-based approach is that one LUT has the capacity to represent more complex functionality than the $n$ neurons it replaces (Carneiro et al. (2019)).

This observation led to approaches that directly learn the LUT/logic gate function with gradient-based methods like ULEEN (Susskind et al. (2023)) and DiffLogicNet (Petersen et al. (2022b)). Building upon these works, the Differentiable Weightless Network (DWN) (Bacellar et al. (2024)) was proposed to enhance training efficiency to support larger LUT sizes and replace the randomized connections used in DiffLogicNet with learnable connections, ultimately achieving greater accuracy with a more compact network.

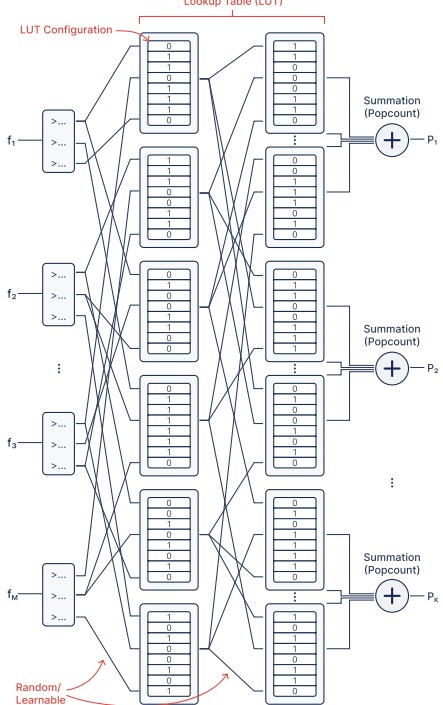

### 2.3.1 DIFFERENTIABLE WEIGHTLESS NETWORK

Fig. 2 illustrated the overall architecture of the Differentiable Weightless Network (DWN). Each of the $M$ input features $(f_1, ..., f_M)$ is first transformed using a thermometer encoding (Bacellar et al. (2022)). The encoded binary output for each feature is then fed into a Look-Up Table (LUT) based on a predefined connection. Each LUT contains a trainable input/output mapping. It accepts $n$ input and produces one binary output. For the final classification into $K$ classes, the final layer LUT outputs are partitioned into $K$ groups. The summation of the binary values is computed within each group. The final prediction is made by performing an argmax operation over these $K$ summation values, selecting the class associated with the group with the highest count of activated LUTs. Both the LUT configuration and connection are randomly initialized and then learned during training.

Figure 2: An overall architecture of Differentiable Weightless Network (DWN) (Bacellar et al. (2024)) for a classification task with $M$ input features and $K$ output class.

## 3 HELUTNET: DESIGN OVERVIEW

This section explains the HeLutNet overall design. We begin by explaining our motivation for selecting the BGV FHE scheme in Sec. 3.1. Then, we explain how we evaluate each LUT and each layer of the network using the integer SIMD computation capability of the BGV FHE scheme in Sec. 3.2. Finally, we explain in Sec. 3.3 a strategy to connect multiple layers of LUTs to form a complete network.

### 3.1 FHE SCHEME SELECTION

The TFHE FHE scheme is well known for fast bootstrapping and efficient logic gate and lookup table implementation through gate bootstrapping and the programmable bootstrapping operation (Chillotti et al. (2021)). Gate bootstrapping and programmable bootstrapping allow the gate and LUT to be chained indefinitely as the computation is performed as part of the bootstrapping operation, which resets the ciphertext's noise level. Unlike ordinary logic circuits, LUT-based neural networks have shallow logic depth and do not benefit from the unlimited computation depth offered by fast TFHE bootstrapping. Instead, we found that a leveled approach combined with a possibility to compute multiple LUTs in a SIMD fashion allows for much faster inference. Appendix A.5 provides a comparison of HeLutNet runtime when implemented with the TFHE vs the BGV scheme.

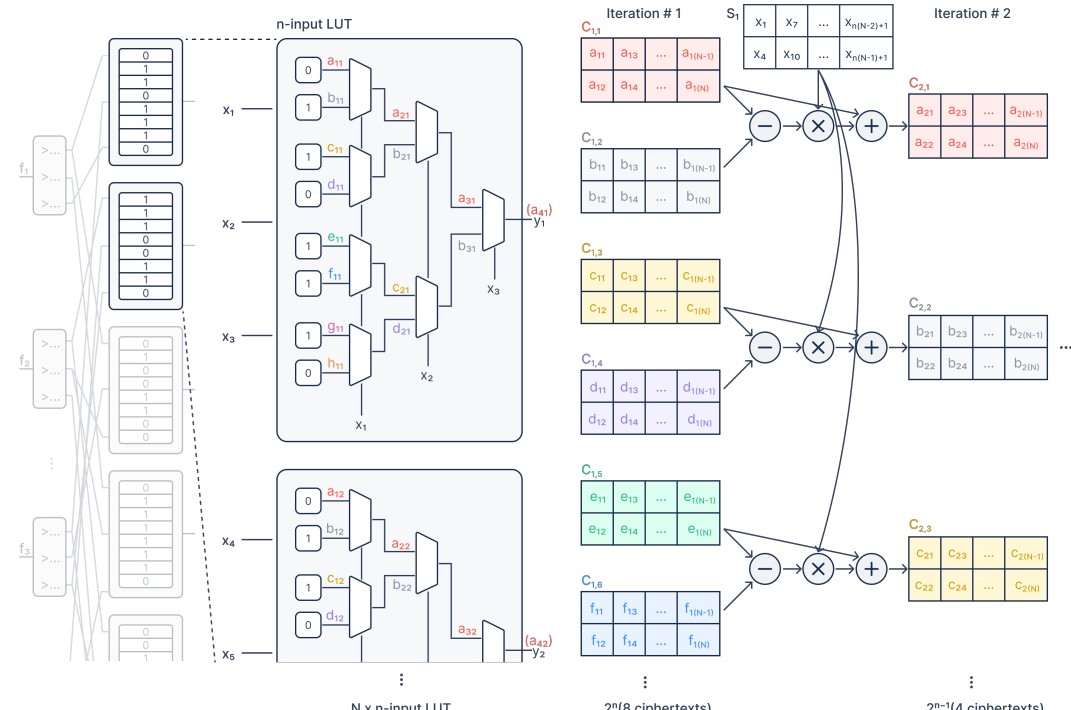

Figure 3: Our strategy for evaluating LUT using a multiplexer tree in a SIMD fashion in BGV FHE scheme by packing several multiplexer inputs and select inputs into the same ciphertext.

## 3.2 LUT IMPLEMENTATION IN BGV FHE SCHEME

In a nutshell, most BGV FHE scheme implementations enable efficient SIMD integer arithmetic operation on a ciphertext containing an encrypted matrix of 2 rows and $P/2$ columns, where $P$(poly_modulus_degree) is the number of slots in the ciphertext.

**Implementing LUT with multiplexer.** To make use of that capability, we use a 2-input multiplexer (Fig. 4) as a building block to build an n-input LUT. A 2-input multiplexer can be implemented with an integer arithmetic expression $F = A + S * (B - A)$ assuming $A$ and $B$ are the two binary inputs, $S$ is a select input ($F = A$ when $S = 0$ and $F = B$ when $S = 1$), and $A, B, S \in \{0, 1\}$. With that, $n$-input LUT can be built using $n$-level tree of 2-input multiplexer by using the LUT's parameter (i.e. its input and output mapping) as inputs to the multiplexer tree ($a_{11} - h_{11}$) and LUT's input as the select input for multiplexer ($x_1 - x_3$). For instance, Fig. 3 shows how 3-input LUT can be implemented with 7 ($2^N - 1$) multiplexer arrange in 3-level with 4, 2, and 1 MUX in each level respectively.

**Packing multiplexer parameter into ciphertext.** To enable SIMD processing, we derive a method to pack multiple multiplexer inputs and select inputs into the same ciphertext to compute $A + S * (B - A)$ in SIMD fashion. Fig. 3 shows an example for the case when the number of LUT input ($n$) = 3 and the number of LUTs in the current layer in the network is $N$. For $n$-input LUT, $n$ iterations are needed to process each level of the multiplexer tree to generate the LUT's output. In the first iteration, $2^n$ ciphertext is used which will be reduced by half in the subsequent iteration until we are left with 1 ciphertext in the last iteration (iteration #$n$). To pack the multiplexer input into ciphertext ($C_{1,1}, C_{1,2}, ..., C_{1,2^n}$), we group the i-th multiplexer input from each LUT in the current layer and pack them into the i-th ciphertext in a column-major order. For instance, $C_{1,1}$ will contain $a_{11}, a_{12}, .., a_{1N}$ (the first input of the first multiplexer in the first level of the multiplexer tree of each LUT in the current layer) and $C_{1,2}$ will contain $b_{11}, b_{12}, .., b_{1N}$ and so on. The significance of choosing column-major instead of row-major ordering is for efficiency and

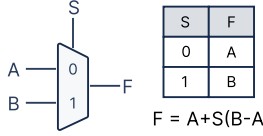

| S | F |
|---|---|
| 0 | A |
| 1 | B |

F = A+S(B-A)

Figure 4: A truth table and integer arithmetic equation of a 2-input multiplexer.

correctness when we feed the output of the first LUT layer to the next layer as will be described in Sec. 3.3. The multiplexer select input $(x_1, x_2, ..., x_{3*N})$ follow the same strategy as the multiplexer input. For instance, $S_1$ will contain $x_1, x_4, ..., x_{n(N-1)+1}$ (the select input of multiplexer in the first level of the multiplexer tree in each LUT) and $S_2$ (used in the second iteration and not shown in the figure) will contain $x_2, x_5, ..., x_{n(N-1)+2}$.

**SIMD multiplexer evaluation.** After packing all parameters into the ciphertext, we pair up all ciphertexts in the current iteration to compute the result of all multiplexers in the current level of the multiplexer tree. For example, follow our example in Fig. 3, in iteration #1 we compute $C_{2,1}$ from $C_{1,1}$ and $C_{1,2}$ and $C_{2,2}$ from $C_{1,3}$ and $C_{1,4}$ and so on, following the equation $C_{2,i} = C_{1,2i-1} + S_1 \times (C_{1,2i} - C_{1,2i-1})$ where $i = 1$ to $2^{n-1}$ using BGV element-wise addition and multiplication operations. Note that the result of the computation follows the same data packing format e.g. $C_{2,1}$ contains $a_{21}, a_{22}, .., a_{2N}$ allowing us to repeat the same step in the subsequent iteration until we get the final ciphertext ($C_{4,1}$).

## 3.3 LUT-BASED NETWORK EVALUATION

From Sec. 3.2, the inputs to the LUT $(x_1, x_2, ..., x_N)$ are used as a select input for the multiplexer tree and they must be packed in an interleaved fashion to enable SIMD processing i.e. the i-th input of each LUT are grouped in ciphertext i-th e.g. $S_1$ should include $x_1, x_4, ..., x_{n(N-1)+1}$, $S_2$ should include $x_2, x_5, ..., x_{n(N-1)+2}$ and so on. While this packing strategy improves the computation efficiency significantly by enabling us to evaluate all LUTs in the same network level at the same time in a single ciphertext operation. It introduces a challenge while the output of the previous layer needs to be used as an input to the next layer as the connection between the two adjacent layer in the LUT network is arbitrary as shown in Fig. 2 and 5. Therefore, we must rearrange value packed in matrix slots of the output ciphertext from the prior layer to have

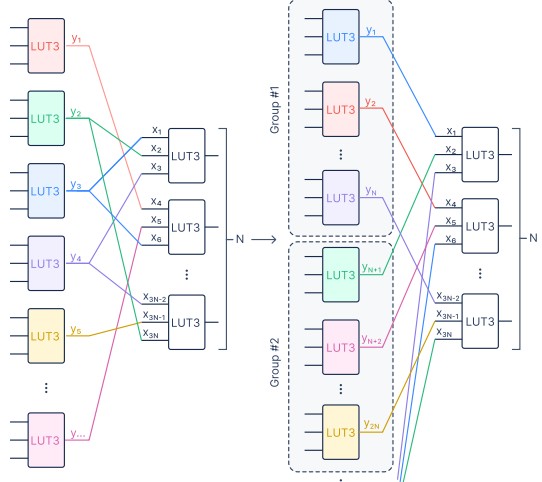

Figure 5: Our LUT rearrangement strategy enables efficient implementation of multi-layer LUT network.

the correct data order to be used in the subsequent layer. For instance, in Fig. 5, the output ciphertext from layer 1 contains (in the point-of-view of the second layer) $x_4(y_1), x_2(y_2), x_1(y_3), ...$ which must be rearranged into $x_1, x_4, ..., x_{n(N-1)+1}$ for $S_1$ and so on. However, rearranging values in the ciphertext is very time-consuming as we must perform masking (element-wise multiplication with all-zero ciphertext with only one element set to 1 at the desired position) and rotating (to shift the desired element to the correct position) operations to generate N temporary ciphertexts. Then, recombine all temporary ciphertexts with an addition operation to generate $S_1$ and repeat for $S_2, ..., S_N$.

To avoid these inefficiencies, we propose a more efficient strategy for LUT-based network implementation by rearranging the LUT in the previous layer during model conversion time (from a PyTorch-trained model to FHE inference code) so that the resulting ciphertext contains each LUT output in the correct order ready to be used by the subsequent layer. Fig. 5 illustrates this approach for 3-input LUT ($n = 3$) with $N$ LUT in the next layer and an arbitrary number of LUT in the current layer. First, we arrange the LUT which produces the output to be used as the first input of each LUT in the next layer as the first $N$ LUT (group #1). Then, we arrange the next $N$ LUT (group #2) to produce output to be used as the second input of each LUT in the next layer. Then, repeat for each input of the LUT ($n$ times) until the number of LUT current layer is $n \times N$.

After rearranging the LUT, we can evaluate each group of LUT separately using the method described in Sec. 3.2. In this case, each group of LUT will produce one output ciphertext, which can be directly used as $S_1 - S_n$. While this *multi ciphertext* approach is straightforward to implement, it necessitates additional rounds of computation for each layer ($n$ rounds per layer). Alternatively, we can utilize the fact that the number of slots in the ciphertext is usually much higher than the number

Table 2: Number of homomorphic operations of HeLutNet's two LUT-network evaluation strategies. $N_1$ and $N_2$ denote the number of LUTs in the first and second layers, respectively, $n$ denotes the number of LUT inputs (LUT size), and $P$ denotes the poly modulus degree (ring degree).

| Impl. Strategy | Num Cipher | Homomorphic Operations | | | | Mult Depth | Min $P$ |
|---|---|---|---|---|---|---|---|
| | | Add/Sub | Mult w/plain | Mult w/cipher | Rotation | | |
| Single | 1 | $3(2^n) - 4$ | $2^n$ | $2^n - 2$ | $n - 1$ | $2n$ | $nN_2$ |
| Multi | $n$ | $n(3(2^n) - 4)$ | $n(2^n)$ | $n(2^n - 2)$ | 0 | $2n$ | $N_2$ |

of LUTs in each level, e.g., 16,384 vs. 2,000. Therefore, we can treat all groups of LUT as a single wider layer of $n \times N$ LUT and pack them continuously in the same ciphertext ($C_{1,1} - C_{1,2^n}$) for evaluation in one round in SIMD fashion. In this *single ciphertext* approach, the packed select input ($S_1 - S_n$) for the next layer can be generated by separating the output ciphertext with $n - 1$ rotation operations, each rotating by $\frac{N}{2}, 2 \times \frac{N}{2}, ..., (n-1) \times \frac{N}{2}$, respectively.

### 3.4 Implementation Selection and Parameters Setting

Tab. 2 details the total number of homomorphic operations for each implementation strategy. The optimal strategy is determined by the number of LUTs in each layer ($N_1$, $N_2$) and the LUT size ($n$). The single-cipher method uses fewer homomorphic operations but requires multiple expensive rotation operations. These rotations require an additional Galois key, increasing key size and communication overhead (see Sec. 5.3). It may also force an increase in the ring degree ($P$) to the next power of two, which slows down computation and enlarges the key. In contrast, the multi-cipher method performs more homomorphic operations but eliminates the need for rotations. It may also enable the use of a smaller ring degree ($P$), leading to a large speedup when $n$ is small. Our experiments indicate that the multi-cipher method is preferred when $n = 2$ and $N_2 < 2^{13}$. Otherwise the single-cipher method is faster. Detailed runtimes are provided in Appendix A.4.

HeLutNet automatically selects the optimal ring degree ($P$) based on $N_1$, $N_2$, and $n$. It defaults to the smallest ring degree supporting enough multiplicative depth, e.g., $2^{13}$ when $n = 2$ and $2^{14}$ when $n = 3 - 6$, and increases if needed based on $N_2$ and $n$. Other FHE parameters are set according to the default values provided by each backend library for the 128-bit security level.

## 4 HeLutNet: Implementation

We implement HeLutNet as a Python library. It accepts PyTorch-trained LUT-based model and generates ready to deploy FHE inference code in C++ or Go leveraging Microsoft SEAL (SEAL) or Lattigo (lat (2024)) library as the backend for low-level FHE computation. While we choose to support model trained using the training method from Bacellar et al. (2024) which is the current state-of-the-art in term of accuracy and training time in the current version of HeLutNet, it should be straightforward to add support for other LUT-based models as they become available.

## 5 Experimental Results

### 5.1 Experimental Setup

**Machine setup:** We evaluated HeLutNet on three machine types: 1) a legacy server (N1 Google Cloud Platform VM with an unspecified Intel Haswell CPU), 2) a modern desktop (Intel Core i7-13700K), and 3) a modern server (C4 Google Cloud Platform VM with an Intel Xeon Platinum 8581C). The modern server configuration is identical to the one used by Orion (Ebel et al. (2025)), the current SOTA FHE-based PPML method. This diverse testing environment allows for a fair latency comparison with all prior works, as comprehensive testing across all different CPUs used in the literature, as detailed in Appendix A.7 is infeasible. For our evaluation, we include HeLutNet implementations based on both Microsoft SEAL (SEAL) and Lattigo (lat (2024)), as these are the most widely used libraries for BGV and CKKS FHE schemes across existing literature.

Table 3: Accuracy and inference time of HeLutNet vs previous FHE-based PPML methods. All latencies are single-sample inferences without batching. Our results are based on GCP C4 VM. The CPU setup for all prior works is detailed in Appendix A.7. [†] indicates that the results are collected by us using the official code release by the authors.

| Dataset | Method | Model | Library | Acc.(%) | Latency (s) | #Thread |
|---|---|---|---|---|---|---|
| MNIST | Our | DWN ($n = 2$) | SEAL/Lattigo | 98.38 | **0.008/0.015** | 1 |
| | Our | DWN ($n = 3$) | SEAL/Lattigo | 98.50 | 0.055/0.126 | 1 |
| | Orion | LoLA | Lattigo | 98.62 | 0.230 | 1 |
| | Orion | MLP | Lattigo | 98.03 | 0.290 | 1 |
| | EVA | LeNet-5 (S) | SEAL | 98.45 | 0.600 | 56 |
| | EVA | LeNet-5 (M) | SEAL | 99.11 | 1.200 | 56 |
| | WiSARDs | WNNs | MOSFHET | 93.76 | 2.184 | 1 |
| | LoLA | CNN | - | 98.95 | 2.200 | 20 |
| | Orion | LeNet-5 | Lattigo | 99.31 | 2.930 | 1 |
| | EVA | LeNet-5 (L) | SEAL | 99.30 | 5.600 | 56 |
| | SHE | CryptoNets | TFHE | 99.54 | 9.300 | 20 |
| | TT-TFHE | TTnet | Concrete | 97.20 | 54.020 | 4 |
| | DSHE | CNN | TFHE | **99.77** | 124.900 | 20 |
| FMNIST | Our | DWN ($n = 2$) | SEAL/Lattigo | 89.75 | **0.008/0.015** | 1 |
| | Our | DWN ($n = 3$) | SEAL/Lattigo | **90.03** | 0.055/0.126 | 1 |
| | Orion[†] | LoLA | Lattigo | 89.28 | 0.190 | 1 |
| | Orion[†] | MLP | Lattigo | 89.90 | 0.280 | 1 |
| HAM10000 | Our | DWN ($n = 2$) | SEAL | **74.49** | **0.008** | 1 |
| | WiSARDs | WNNs | MOSFHET | 69.85 | 6.74 | 1 |
| Adult | Our | DWN ($n = 2$) | SEAL | **87.10** | **0.008** | 1 |
| | TT-TFHE | TTnet | Concrete | 85.30 | 89.1 | 4 |
| Cancer | Our | DWN ($n = 2$) | SEAL | **98.25** | **0.008** | 1 |
| | WiSARDs | WNNs | MOSFHET | 97.30 | 0.026 | 1 |
| | HBDT | Decision Tree | | 93.64 | 1.650 | 1 |
| | TT-TFHE | TTnet | Concrete | 97.10 | 1.9 | 4 |
| Diabetes | Our | DWN ($n = 2$) | SEAL | **59.98** | **0.008** | 1 |
| | TT-TFHE | TTnet | Concrete | 57.00 | 18.16 | 4 |

**Datasets:** Our evaluation consists of 20 datasets across four categories: vision (MNIST, Fashion-MNIST, HAM10000), speech (Qualcomm Keyword Speech Dataset (QKSD), Free Spoken Digit (FSDD)), health (PTB diagnostic ECG (PTB), UCI Human Activity Recognition (UCIHAR), Cancer, Diabetes), and tabular data (Adult, and 10 datasets from Bacellar et al. (2024). The detail description of each dataset and the preprocessing step is provided in Tab. 7 in Appendix A.1.

## 5.2 ACCURACY AND INFERENCE TIME

Tab. 3 presents the accuracy and single-sample inference time of HeLutNet compared to prior FHE-based PPML work. For vision datasets, specifically when compared to Orion (Ebel et al. (2025)), the current SOTA FHE-based DNNs method, HeLutNet achieves competitive accuracy on both MNIST (98.38% vs 98.62%) and Fashion-MNIST (89.75% vs 89.28%). Crucially, HeLutNet significantly reduces the single-sample inference runtime: from 230 ms to 8.3 ms (a 27.71x speedup) on MNIST and from 190 ms to 8.3 ms (a 22.89x speedup) on Fashion-MNIST. When compared to the similar weightless neural network (WNN) methods such as WiSARDs and TT-TFHE, our approach consistently improves accuracy across all evaluated vision, tabular, and health datasets, including MNIST (98.38% vs 97.20%), HAM10000 (74.49% vs 69.85%), Adult (87.10% vs 85.30%), Cancer (98.25% vs 97.30%), and Diabetes (58.98% vs 57.00%). These results demonstrate HeLutNet superiority to existing WNNs in terms of both accuracy and inference runtime.

Tab. 4 provides comparisons of inference time and accuracy for datasets without existing FHE baselines. For these datasets, we trained decision tree (DT), support vector machine (SVM), and extreme gradient boosting (XGBoost) models using Concrete-ML (Zama (2022)), a SOTA open-

Table 4: Accuracy and inference time of HeLutNet vs Concrete-ML Library (Zama (2022)) on several speech, health, and tabular datasets. All latencies are single-sample inferences without batching.

| Dataset | Our | | Decision Tree | | SVM | | XGBoost | |
|---|---|---|---|---|---|---|---|---|
| | Acc(%) | Time(s) | Acc(%) | Time(s) | Acc(%) | Time(s) | Acc(%) | Time(s) |
| QKSD | **98.34** | **0.008** | 78.42 | 3.663 | 93.62 | 0.082 | 97.73 | 118.405 |
| FSDD | **93.83** | **0.008** | 56.67 | 3.238 | 90.33 | 0.089 | 91.33 | 295.702 |
| PTB | 97.29 | **0.008** | 85.54 | 5.604 | 83.13 | 0.018 | **98.08** | 85.966 |
| UCIHAR | 95.18 | **0.008** | 82.05 | 0.981 | **96.40** | 0.055 | 94.77 | 79.539 |
| Phoneme | 89.55 | 0.055 | 82.89 | 5.240 | 77.15 | **0.001** | **89.82** | 126.281 |
| Skin-seg | **99.94** | 0.055 | 98.91 | 5.224 | 92.38 | **0.001** | 99.94 | 129.109 |
| Australian | 86.23 | 0.008 | 84.06 | 0.439 | 81.16 | **0.002** | **88.41** | 17.660 |
| Nomao | **97.24** | 0.055 | 94.07 | 6.757 | 94.46 | **0.012** | 97.03 | 207.852 |
| Segment | **97.62** | 0.008 | 89.39 | 0.722 | 71.86 | **0.003** | 97.62 | 147.922 |
| Miniboone | **94.78** | 0.055 | 88.41 | 6.309 | 71.73 | **0.004** | 90.93 | 230.878 |
| Christine | **75.37** | **0.055** | 69.00 | 2.009 | 69.93 | 0.158 | 73.43 | 16.839 |
| Jasmine | **82.24** | 0.055 | 78.22 | 0.885 | 76.55 | **0.014** | 81.07 | 78.950 |
| Sylvine | **96.10** | 0.055 | 92.39 | 2.654 | 73.27 | **0.003** | 94.15 | 86.751 |
| Blood | **80.00** | 0.055 | 79.33 | 1.518 | 76.67 | **0.001** | 78.67 | 11.311 |

Table 5: Key sizes, ciphertext sizes, total communication, and total memory usage of HeLutNet during inference on a single input sample. Results are collected using the HeLutNet C++ implementation based on Microsoft SEAL. The BGV scheme is used when $n = 2$-5. For $n = 6$, the BFV scheme is used due to its higher noise budget at the expense of computation overhead.

| Model Configuration | Key Size | | Ciphertext Size | | Total Communication | Total Memory |
|---|---|---|---|---|---|---|
| | Relin | Galois | Input | Output | | |
| $n=2$ (Multi), $P=2^{13}$ | 1.1 MB | - | 865.6 kB | 214.3 kB | 2.18 MB | 38 MB |
| $n=2$ (Single), $P=2^{14}$ | 8.2 MB | 8.2 MB | 1.82 MB | 448.3 kB | 18.67 MB | 145 MB |
| $n=3$ (Single), $P=2^{14}$ | 8.2 MB | 16.5 MB | 2.74 MB | 903.0 kB | 28.34 MB | 196 MB |
| $n=4$ (Single), $P=2^{14}$ | 8.2 MB | 24.7 MB | 3.65 MB | 903.0 kB | 37.46 MB | 323 MB |
| $n=5$ (Single), $P=2^{14}$ | 8.2 MB | 32.9 MB | 4.57 MB | 903.0 kB | 46.57 MB | 672 MB |
| $n=6$ (Single), $P=2^{14}$ | 8.2 MB | 41.2 MB | 5.48 MB | 448.2 kB | 55.33 MB | 1,071 MB |

source FHE implementation, to match the best-known unencrypted inference accuracy reported in Bacellar et al. (2024) and Narkthong et al. (2024). For the speech datasets (QKSD and FSDD), HeLutNet achieves superior accuracy while performing each inference in only 8.3 ms, demonstrating the potential as the first real-time FHE-based speech/command recognition system, e.g., in smart speakers. On health datasets (PTB and UCIHAR), HeLutNet's accuracy is slightly lower than that of SVM and XGBoost, but its inference speed remains significantly faster. For the tabular datasets, HeLutNet achieves the highest accuracy across all datasets except Phoneme and Australian. While SVM is the fastest overall, it suffers from significantly lower accuracy. When accuracy is considered, the HeLutNet runtime is much faster than the similarly accurate XGBoost and DT models.

## 5.3 MEMORY AND COMMUNICATION OVERHEAD

Tab. 5 details the key sizes, ciphertext sizes, total communication (between client and server), and total memory usage of HeLutNet during inference on a single input sample. Key and ciphertext sizes were collected by measuring the data size after serialization, which represents the communication overhead. Total memory required during inference was recorded using gperftools (gperftools (2025)). All results are based on our C++ implementation based on Microsoft SEAL. Tab. 6 presents a comparison of these metrics against existing FHE-based PPML work. Results show that HeLutNet achieves significantly lower key size, ciphertext size, total communication size, and total memory usage compared to all prior works evaluated across all datasets. Note that many methods in Tab. 3 did not report these metrics and they were excluded from Tab. 6.

Table 6: Key sizes, ciphertext sizes, total communication, and total memory usage of HeLutNet vs prior FHE-based PPML methods across various datasets. "-" indicates that the data was not reported.

| Dataset | Method | Key Size | Cipher Size | Total Comm. | Total Mem |
|---------|--------|----------|-------------|-------------|-----------|
| MNIST | Our ($n = 2$) | 1.1 MB | 1.08MB | 2.18 MB | 38 MB |
| | TT-TFHE (Benamira et al. (2025)) | 766.9MB | 102.5 MB | 869.4 MB | 53.7MB |
| | SHE (Lou & Jiang (2019)) | - | 123 MB | - | - |
| | DiNN (Lou & Jiang (2019)) | - | 66 MB | - | - |
| Adult | Our ($n = 2$) | 1.1 MB | 1.08MB | 2.18 MB | 38 MB |
| | TT-TFHE (Benamira et al. (2025)) | 1.56 GB | 13 MB | 1.57 GB | 15 GB |
| Cancer | Our ($n = 2$) | 1.1 MB | 1.08MB | 2.18 MB | 38 MB |
| | TT-TFHE (Benamira et al. (2025)) | 168.98 MB | 1.7 MB | 170.68 MB | - |
| Diabetes | Our ($n = 2$) | 1.1 MB | 1.08MB | 2.18 MB | 38 MB |
| | TT-TFHE (Benamira et al. (2025)) | 1.53 GB | 37.4 MB | 1.57GB | 6.9 GB |

## 6 DISCUSSION

**Scalability and Practicality of HeLutNet.** At present, LUT-based methods, including HeLutNet, are not yet applicable to large datasets such as CIFAR-10 or ImageNet. Nevertheless, our evaluation demonstrates their immediate practicality for numerous real-world datasets, enabling the first practical real-time FHE-based PPML inference system. For instance, our solution efficiently performs real-time inference on a single CPU core for small vision datasets (MNIST, Fashion-MNIST, and HAM10000), processes ECG data for heartbeat pattern detection (PTB), analyzes motion sensor data for human activity recognition (UCIHAR), and handles speech signals for speech recognition (QKS and FSDD). This proves the viability of our approach for a range of privacy-sensitive applications.

**Comparison with Secure Multi-party Computation (MPC).** While Secure Multi-party Computation (MPC) protocols are generally less computationally intensive and exhibit lower latency than FHE, they carry inherent limitations. MPC can potentially leak information and requires multiple parties to ensure security, which also introduces additional communication overhead. In contrast, we propose an FHE-based inference library that provides a stronger security guarantee while achieving lower latency than state-of-the-art MPC systems (see Appendix. A.6). This makes our FHE-based approach a compelling solution that surpasses the performance of existing MPC methods while offering enhanced security.

## 7 CONCLUSION

In this paper, we introduced HeLutNet, a novel approach that leverages a look-up table (LUT)-based ML model to enable extremely fast, FHE-based privacy-preserving inference (PPML). HeLutNet converts a PyTorch-trained, LUT-based model into an efficient FHE program, implementing LUTs using the BGV scheme's SIMD capabilities and incorporating novel packing and LUT reordering strategies to minimize the number of homomorphic operations required. Our evaluation on 20 datasets across four categories (vision, speech, health, and tabular data) demonstrates the practicality and superiority of our approach both in terms of accuracy and inference latency. Specifically, HeLutNet significantly reduces the single-sample inference time on a single CPU core to as low as 8.3 ms across various datasets, representing a remarkable 27.71x speedup over the current SOTA FHE deep neural network methods. Crucially, HeLutNet achieves this performance while dramatically improving resource efficiency, demonstrating significantly smaller key size, ciphertext size, communication overhead, and memory usage compared to prior FHE-based and MPC-based privacy-preserving works. These results establish HeLutNet as the first practical, real-time FHE-based PPML inference framework on commodity hardware, paving the way for the widespread deployment of privacy-preserving machine learning services.

## REPRODUCIBILITY STATEMENT

The code of HeLutNet and all benchmarks included in this paper are open-sourced and available at `https://anonymous.4open.science/r/HeLutNet-2F56/`.

## LLM USAGE DISCLOSURE

An LLM tool was used to proofread this paper for grammatical and stylistic errors.

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

# A  APPENDIX

## A.1  LIST OF ALL DATASETS USED IN THIS PAPER

We use the standard train/test split for each dataset when available. For datasets that don't provide separate train/test dataset files, we divide the data into an 80% training set and a 20% testing set. For speech datasets, including the Qualcomm Keyword Speech Dataset (QKSD) and the Free Spoken Digit Dataset (FSDD), we follow the preprocessing steps outlined in Narkthong et al. (2024).

Table 7: List of all dataset used in this paper.

| | Dataset | Source | Description |
|---|---|---|---|
| **Vision** | MNIST | Deng (2012) | Handwritten digit images dataset |
| | Fashion-MNIST | Xiao et al. (2017) | Fashion products images dataset |
| | HAM10000 | Tschandl et al. (2018) | Dermatoscopic image data for pigmented skin lesion classification and diagnosis |
| **Speech** | QKSD | Kim et al. (2019) | Qualcomm Keyword Speech Dataset |
| | FSDD | Jackson et al. (2018) | Audio recordings of spoken digits |
| **Health** | PTB | Kachuee et al. (2018) | ECG recordings for cardiac diagnosis |
| | UCIHAR | Anguita et al. (2013) | UCI Human Activity Recognition |
| | Cancer | Wolberg & Street (1993) | FNA-derived feature data for classifying malignant and benign tumors |
| | Diabetes | Clore & Strack (2014) | Clinical records for predicting 30-day readmission of diabetes patients |
| **Tabular** | Phoneme | Cappel (1993) | Audio features for phoneme classification |
| | Skin-seg | Bhatt & Dhall (2009) | Pixel data for skin vs. non-skin segmentation |
| | Australian | Quinlan (1987) | Credit approval data with categorical and numerical features |
| | Nomao | Candillier & Lemaire (2012) | Noisy and sparse features for binary classification |
| | Segment | UCI (1990) | Image region data for object segmentation/classification |
| | Miniboone | Roe (2005) | Particle physics data for signal/background classification |
| | Christine | CodaLab (2016) | Synthetic dataset from AutoML competition |
| | Jasmine | CodaLab (2016) | Synthetic dataset from AutoML competition |
| | Sylvine | CodaLab (2016) | Synthetic dataset from AutoML competition |
| | Blood | Yeh (2008) | Blood donation data for donor return prediction |
| | Adult | Becker & Kohavi (1996) | Census demographic data for predicting whether income exceeds $50K |

A.2 HeLutNet: model architecture, accuracy, and inference time

Tab. 8 shows the model training parameter, including the number of bits used for binarization and quantization with distributive thermometer encoding (Bacellar et al. (2022)), the number of LUT inputs (LUT size), and the number of LUTs in each layer (model dimension). We adopt the DWN training code from Bacellar et al. (2024) and use the HeLutNet tool to export FHE inference code in C++ based on the Microsoft SEAL library to measure the inference latency. Latency on other machine configurations can be determined using information in Appendix A.4.

Table 8: Accuracy and inference time of HeLutNet across various vision, speech, healthcare, and tabular datasets. Runtime are collected on our C++ implementation based on Microsoft SEAL on a C4 Google Cloud Platform instance with Intel Xeon Platinum 8581C using single CPU core.

| | Dataset | Binarize #bits | Input Dimension | LUT Size | Model Dimension | Accuracy (%) | Latency (ms) |
|---|---|---|---|---|---|---|---|
| Vision | MNIST | 3 | 3x28x28 | 2 | 6000, 6000 | 98.38 | 8.30 |
| | | | | 3 | 6000, 3000 | 98.50 | 8.30 |
| | Fashion-MNIST | 7 | 7x28x28 | 2 | 8000, 8000 | 89.75 | 8.30 |
| | | | | 3 | 5000, 2500 | 90.03 | 82.84 |
| | HAM10000 | 20 | 20x3x28x28 | 2 | 7000,3500 | 74.49 | 8.30 |
| Speech | QKSD | 3 | 3x810 | 2 | 2000,2000 | 98.34 | 8.30 |
| | FSDD | 3 | 3x800 | 2 | 4000,4000 | 93.83 | 8.30 |
| Health | PTB | 3 | 3x187 | 2 | 4000,4000 | 97.29 | 8.30 |
| | UCIHAR | 3 | 3x561 | 2 | 4200,4200 | 95.18 | 8.30 |
| | Cancer | 100 | 100x3 | 2 | 8,8 | 98.25 | 8.30 |
| | Diabetes | 100 | 100x47 | 2 | 3600,3000 | 59.98 | 8.30 |
| Tabular | Phoneme | 200 | 200x5 | 3 | 5000, 5000 | 89.55 | 82.84 |
| | Skin-seg | 200 | 200x3 | 3 | 4000, 4000 | 99.94 | 82.84 |
| | Australian | 200 | 200x14 | 2 | 5000, 5000 | 86.23 | 8.30 |
| | Nomao | 200 | 200x118 | 3 | 4000, 4000 | 97.24 | 82.84 |
| | Segment | 200 | 200x19 | 2 | 4200, 4200 | 97.62 | 8.30 |
| | Miniboone | 200 | 200x50 | 3 | 4000, 4000 | 94.78 | 82.84 |
| | Christine | 20 | 20x1636 | 3 | 4000, 4000 | 75.37 | 82.84 |
| | Jasmine | 200 | 200x144 | 3 | 4000, 4000 | 82.24 | 82.84 |
| | Sylvine | 200 | 200x20 | 3 | 4000, 4000 | 96.10 | 82.84 |
| | Blood | 200 | 200x4 | 3 | 4000, 4000 | 80.00 | 82.84 |
| | Adult | 150 | 150x14 | 2 | 2000,2000 | 87.10 | 8.30 |

### A.3 CONCRETE-ML: ACCURACY AND INFERENCE TIME

Tab. 9 shows the accuracy and runtime using 1, 8, and 16 CPU cores on 14 datasets based on the Concrete-ML (Zama (2022)) library, a SOTA implementation of FHE-based privacy-preserving decision tree (DT), support vector machine (SVM), and XGBoost based on the TFHE FHE scheme. For each dataset, we train three machine learning models: DT, SVM, and XGBoost using quantization-aware training and grid search to find the best hyperparameter. The code to train and perform the benchmark is available in the HeLutNet's GitHub repository.

Table 9: Accuracy and inference time of fully homomorphic inference with Concrete-ML (Zama (2022)) library across various models, datasets, number of CPU cores, and number of bits to use for quantization. Runtime are collected on a C4 Google Cloud Platform instance with Intel Xeon Platinum 8581C @2.3GHz. (C = CPU Cores)

| Dataset | Model | 4 Bit | | | | 8 Bit | | | |
| | | Acc (%) | Time (s) | | | Acc (%) | Time (s) | | |
| | | | 1 C | 8 C | 16 C | | 1 C | 8 C | 16 C |
|---|---|---|---|---|---|---|---|---|---|
| PTB | DT | 89.39 | 0.72 | 0.21 | 0.17 | 88.96 | 1.33 | 0.33 | 0.25 |
| | XGB | 96.97 | 111.98 | 26.56 | 16.73 | **97.62** | 147.92 | 31.34 | 20.80 |
| QKSD | SVM | 88.13 | 0.06 | 0.06 | 0.06 | 93.62 | 0.08 | 0.08 | 0.08 |
| | XGB | **97.73** | 118.41 | 16.87 | 9.28 | 97.66 | 206.57 | 32.85 | 17.09 |
| FSDD | SVM | 58.67 | 0.06 | 0.06 | 0.06 | 90.33 | 0.09 | 0.08 | 0.09 |
| | XGB | 90.00 | 157.94 | 20.16 | 10.34 | **91.33** | 295.702 | 40.69 | 19.28 |
| UCIHAR | SVM | 34.20 | 0.04 | 0.04 | 0.04 | **96.40** | 0.05 | 0.05 | 0.05 |
| | DT | 80.56 | 0.59 | 0.22 | 0.21 | 82.05 | 0.98 | 0.31 | 0.30 |
| | XGB | 94.57 | 40.63 | 5.45 | 2.96 | 94.77 | 79.54 | 10.65 | 5.13 |
| Phoneme | DT | 82.05 | 3.25 | 0.55 | 0.40 | 82.89 | 5.24 | 0.81 | 0.53 |
| | XGB | 86.31 | 77.13 | 12.41 | 6.47 | **89.82** | 126.28 | 24.49 | 12.53 |
| Skin-seg | DT | 98.77 | 2.14 | 0.40 | 0.30 | 98.91 | 5.22 | 0.77 | 0.53 |
| | XGB | 99.71 | 77.84 | 8.84 | 4.78 | **99.94** | 129.11 | 20.53 | 11.27 |
| Australian | DT | 84.06 | 0.28 | 0.14 | 0.13 | 84.06 | 0.44 | 0.20 | 0.20 |
| | XGB | **88.41** | 17.66 | 2.38 | 1.44 | 87.68 | 33.14 | 4.34 | 2.41 |
| Nomao | SWM | 93.52 | 0.01 | 0.01 | 0.01 | 94.46 | 0.01 | 0.01 | 0.01 |
| | DT | 94.04 | 4.49 | 0.56 | 0.41 | 94.07 | 6.76 | 1.01 | 0.62 |
| | XGB | 96.56 | 98.53 | 15.72 | 8.30 | **97.03** | 207.85 | 28.89 | 15.33 |
| Segment | DT | 89.39 | 0.72 | 0.21 | 0.17 | 88.96 | 1.33 | 0.33 | 0.25 |
| | XGB | 96.97 | 111.98 | 26.56 | 16.73 | **97.62** | 147.92 | 31.34 | 20.80 |
| Miniboone | DT | 84.17 | 3.88 | 0.62 | 0.41 | 88.41 | 6.31 | 0.99 | 0.62 |
| | XGB | 84.90 | 92.24 | 17.91 | 9.03 | **90.93** | 230.88 | 32.72 | 17.55 |
| Christine | SWM | 64.39 | 0.11 | 0.11 | 0.11 | 69.93 | 0.16 | 0.16 | 0.16 |
| | DT | 66.70 | 1.11 | 0.39 | 0.33 | 69.00 | 2.01 | 0.59 | 0.49 |
| | XGB | **73.43** | 16.84 | 2.34 | 1.35 | 72.69 | 31.16 | 4.26 | 2.37 |
| Jasmine | DT | 78.22 | 0.88 | 0.23 | 0.18 | 77.22 | 1.68 | 0.38 | 0.27 |
| | XGB | 80.23 | 45.84 | 6.32 | 3.40 | **81.07** | 78.95 | 11.00 | 5.68 |
| Sylvine | DT | 91.02 | 1.33 | 0.26 | 0.22 | 92.39 | 2.65 | 0.48 | 0.38 |
| | XGB | 92.78 | 47.17 | 7.37 | 3.80 | **94.15** | 86.74 | 12.76 | 7.05 |
| Blood | DT | **79.33** | 0.90 | 0.21 | 0.17 | 79.33 | 1.52 | 0.33 | 0.24 |
| | XGB | 78.00 | 6.13 | 0.99 | 0.55 | 78.67 | 11.31 | 1.72 | 1.02 |

## A.4 HeLutNet's runtime on different machine setup and FHE libraries

Tab. 10 - 12 shows the inference time of our single- and multi-cipher C++ implementation based on Microsoft SEAL across various LUT sizes ($n$) and number of CPU cores (1 core - 8 cores). We set the ring (poly modulus) degree ($P$) to a common value for each implementation, which is $2^{13}$ for multi-cipher implementation with $n = 2$ and $2^{14}$ otherwise. Therefore, these run times are valid for any two-layer models with $N_2 \leq P/n$, where $N_2$ is the number of LUTs in the second layer.

Table 10: Inference time of HeLutNet single- and multi-cipher C++ implementation based on Microsoft SEAL across various LUT sizes and numbers of CPU cores. Runtimes are collected on a modern server setup (Google Cloud Platform C4 VM with Intel Xeon Platinum 8581C @2.3GHz).

| LUT Size | Single-cipher Inference Time (ms) | | | | Multi-cipher Inference Time (ms) | | | |
|---|---|---|---|---|---|---|---|---|
| | 1 C | 2 C | 4 C | 8 C | 1 C | 2 C | 4 C | 8 C |
| 2 | 13.26 | 10.93 | - | - | 8.32 | 7.06 | - | - |
| 3 | 55.04 | 32.52 | 29.28 | - | 127.00 | 79.83 | 71.24 | - |
| 4 | 161.68 | 92.46 | 63.36 | 59.27 | 404.52 | 237.72 | 167.29 | 155.96 |
| 5 | 1,129.33 | 593.14 | 361.69 | 279.08 | 3,819.17 | 1,997.33 | 1,228.40 | 944.27 |
| 6 | 2,093.95 | 1,073.91 | 601.55 | 392.94 | 8,785.52 | 4,565.64 | 2,573.56 | 1,688.93 |

Table 11: Inference time of HeLutNet single- and multi-cipher C++ implementation based on Microsoft SEAL across various LUT sizes and numbers of CPU cores. Runtimes are collected on a legacy server setup (Google Cloud Platform N1 VM with unspecified Intel Haswell CPU).

| LUT Size | Single-cipher Inference Time (ms) | | | | Multi-cipher Inference Time (ms) | | | |
|---|---|---|---|---|---|---|---|---|
| | 1 C | 2 C | 4 C | 8 C | 1 C | 2 C | 4 C | 8 C |
| 2 | 57.89 | 48.95 | - | - | 36.86 | - | - | - |
| 3 | 240.30 | 143.47 | 129.80 | - | 534.95 | 330.91 | 298.40 | - |
| 4 | 682.87 | 399.81 | 271.20 | 254.26 | 1,714.08 | 988.41 | 695.34 | 649.69 |
| 5 | 3,367.82 | 1,764.36 | 1,045.61 | 783.29 | 10,984.11 | 5,769.08 | 3,434.05 | 2,571.81 |
| 6 | 6,235.60 | 3,208.88 | 1,746.97 | 1,097.61 | 25,833.85 | 13,304.85 | 7,206.59 | 4,571.05 |

Table 12: Inference time of HeLutNet single- and multi-cipher C++ implementation based on Microsoft SEAL across various LUT sizes and numbers of CPU cores. Runtimes are collected on a modern desktop setup (Intel Core i7-13700K CPU).

| LUT Size | Single-cipher Inference Time (ms) | | | | Multi-cipher Inference Time (ms) | | | |
|---|---|---|---|---|---|---|---|---|
| | 1 C | 2 C | 4 C | 8 C | 1 C | 2 C | 4 C | 8 C |
| 2 | 23.09 | 20.66 | - | - | 14.24 | 14.06 | - | - |
| 3 | 89.92 | 59.11 | 52.12 | - | 195.59 | 127.71 | 116.59 | - |
| 4 | 257.66 | 154.95 | 112.45 | 104.80 | 656.85 | 381.77 | 274.64 | 263.18 |
| 5 | 1,093.43 | 604.18 | 373.16 | 307.96 | 3,652.77 | 1,998.85 | 1,227.16 | 978.41 |
| 6 | 2,107.13 | 1,103.66 | 612.76 | 447.61 | 8,633.57 | 4,570.99 | 2,582.45 | 1,805.96 |

To allow direct comparison with some recent SOTA works which are implemented in Go using the Lattigo library, we present the inference time of HeLutNet single- and multi-cipher Go implementation based on Lattigo across common LUT sizes and numbers of CPU cores in Tab. 13.

Table 13: Inference time of HeLutNet single- and multi-cipher Go implementation based on Lattigo across common LUT sizes and numbers of CPU cores. Runtimes are collected on a modern server setup (Google Cloud Platform C4 VM with Intel Xeon Platinum 8581C @2.3GHz).

| LUT Size | Time (ms) | |
| --- | --- | --- |
| | Single | Multi |
| 2 | 53.60 | 15.60 |
| 3 | 125.56 | 218.00 |

## A.5 HeLutNet BGV-based vs TFHE-based Implementation

Tab. 14 compares the runtime of HeLutNet BGV-based and TFHE-based implementation on several datasets with different model dimensions. In our TFHE-based implementation, we perform $(n-1)N_2$ add, $(n-1)N_2$ multiply with plaintext, and $N_1 + N_2$ PBS ($n$-LUT evaluation) operations, where $n$ is the LUT size and $N_1$ and $N_2$ are the numbers of LUTs in the first and second layers, respectively. For BGV implementation, the number of operations is detailed in Tab. 2.

Table 14: HeLutNET model dimension and inference time across various datasets, LUT Size, Schemes and number of threads and cores. All runtimes are collected on a modern server setup (Google Cloud Platform C4 VM with Intel Xeon Platinum 8581C @2.3GHz).

| Dataset | LUT Size | Model Dimension | HeLutNet (BGV) | HeLutNet (TFHE) | | | |
| --- | --- | --- | --- | --- | --- | --- | --- |
| | | | 1 C | 1 C | 16 C | 32 C | 48 C |
| MNIST | 2 | 6000,6000 | 0.008s | 86.21s | 5.39s | 2.71s | 1.94s |
| | 3 | 6000,3000 | 0.055s | 76.60s | 4.63s | 2.34s | 1.87s |
| FMNIST | 2 | 8000,8000 | 0.008s | 113.84s | 7.17s | 3.60s | 2.58s |
| | 3 | 5000,2500 | 0.055s | 64.07s | 3.87s | 1.97s | 1.43s |
| PTB | 2 | 4000,4000 | 0.008s | 57.81s | 3.59s | 1.81s | 1.30s |
| HAR | 2 | 4200,4200 | 0.008s | 59.85s | 3.77s | 1.93s | 1.38s |
| QKSD | 2 | 2000,2000 | 0.008s | 28.94s | 1.80s | 0.92s | 0.65s |
| FSDD | 2 | 4000,4000 | 0.008s | 57.81s | 3.59s | 1.81s | 1.30s |

## A.6 Communication overhead of HeLutNet vs. secure multi-party computation

Tab. 15 presents a comparison of runtime and communication overhead between HeLutNet and existing MPC-based PPML approaches on the MNIST dataset. While our method achieves slightly lower accuracy, it achieves a runtime that is approximately three times faster than SWIFT (Fu et al. (2025)) and reduces communication cost by a factor of 4.34 compared to MD-ML (Yuan et al. (2024)).

Table 15: HeLutNet vs previous secure multi-party computation (MPC) work in term of accuracy, inference time and communication cost. "-" indicates that the data was not reported.

| Method | Model | Acc. (%) | LAN Inference Time (ms) | Comm. (MB) | References |
| --- | --- | --- | --- | --- | --- |
| Our | DWN (n=2) | 98.38 | 8.3 | 2.18 | |
| SWIFT | CNN | - | 24.14 | - | Fu et al. (2025) |
| Cheetah | CNN | - | 42.88 | - | Fu et al. (2025) |
| MD-ML | LeNet | 98.43 | 80 | 9.46 | Yuan et al. (2024) |
| SPDZ | LeNet | - | 274 | 18.99 | Yuan et al. (2024) |
| Chameleon | CNN | 99.00 | 2240 | 10.5 | Riazi et al. (2018) |

## A.7 BENCHMARK MACHINE SETUP OF HELUTNET AND EXISTING WORKS

Tab. 16 summarizes the CPU models employed in prior FHE-based PPML work. Since comprehensive benchmarking across all previous CPU platforms is infeasible, we evaluate HeLutNet on three distinct machine setups (detailed in Section 5.1). This provides readers with a basis to extrapolate performance to their specific CPU architecture.

Table 16: Summary of CPU model used during evaluation of prior FHE-based PPML work.

| Method | CPU |
|---|---|
| Orion (Ebel et al. (2025)) | Google Cloud Platform C4 VM (Intel Xeon Platinum 8581C @2.3GHz) |
| WiSARDs (Neumann et al. (2025)) | Intel Core i7-12700K @5.0GHz |
| TT-TFHE (Benamira et al. (2025)) | Intel Core i7-8650U @1.9GHz |
| HBDT Shin et al. (2024) | AMD Ryzen 5950X @3.4GHz |
| EVA (Dathathri et al. (2020)) | Intel Xeon Gold 5120 @2.2GHz |
| (D)SHE (Lou & Jiang (2019)) | Intel Xeon E7-4850 @2.0 GHz |
| LoLA (Brutzkus et al. (2019)) | Azure B8ms VM (Unspecified CPU model) |
| FHE-DiNN (Bourse et al. (2018)) | Intel Core i7-4720HQ @2.60 GHz |

