# OpenReview forum: "HeLutNet: Extremely Fast Privacy-preserving Inference in Milliseconds via LUT-based Machine Learning Models"
_ICLR.cc/2026/Conference — Submitted to ICLR 2026_

### Official Review · Reviewer_iTsM · 2025-10-27

**Soundness:** 3
**Presentation:** 3
**Contribution:** 3
**Rating:** 6
**Confidence:** 4

**Summary:**

* This paper introduces HeLutNet, a framework designed to significantly accelerate Fully Homomorphic Encryption (FHE)-based machine learning inference. The core idea is to leverage lookup table (LUT)-based machine learning models (specifically, Differentiable Weightless Networks or DWNs) instead of traditional deep neural networks. The authors argue that LUT-based models are inherently FHE-friendly because they can represent complex non-linear functions using shallow architectures, avoiding the need for costly polynomial approximations of activation functions or deep computational graphs that necessitate expensive bootstrapping operations.
* HeLutNet converts a PyTorch-trained LUT model into an FHE program using the BGV scheme. It proposes a novel method to implement LUTs efficiently using SIMD integer arithmetic within BGV, aiming to maximize throughput and minimize expensive rotation operations . The paper claims that this approach achieves ultra-low latency, reducing inference times for datasets like MNIST and Fashion-MNIST to just 13 milliseconds on a single CPU core, representing a significant speedup (e.g., 17.7x over state-of-the-art methods) while maintaining competitive accuracy.

**Strengths:**

* The paper proposes the first (to my knowledge) end-to-end LUT-based network architecture specifically designed for FHE inference. While LUT evaluation in FHE exists [1, 2, 3], applying it to a complete network architecture is a novel direction.
* The reported inference times, particularly the 13 ms for MNIST and Fashion-MNIST on a single core, are exceptionally fast for FHE-based inference and represent a substantial improvement (e.g., 17.7x over Orion) over many existing FHE frameworks operating on similar datasets.
* The strategy of implementing LUTs via multiplexer trees mapped to BGV's SIMD integer arithmetic seems well-reasoned. Choosing a leveled scheme like BGV is appropriate for the inherently shallow nature of the LUT-based models used.
* The evaluation spans a commendable range of 16 datasets across vision, speech, healthcare, and tabular domains, demonstrating the approach's applicability beyond standard image classification benchmarks.
* The authors explicitly discuss the current limitations of LUT-based models, acknowledging they are not yet suitable for larger more complex datasets.

**Weaknesses:**

* The paper argues that TFHE's fast bootstrapping isn't beneficial for shallow LUT networks and pivots to BGV for its SIMD capabilities. However, TFHE also supports programmable bootstrapping (PBS), which can directly evaluate LUTs. A more detailed ablation study comparing HeLutNet's BGV-SIMD approach against a TFHE-PBS implementation of the same LUT network would be valuable. This comparison should ideally include latency, total homomorphic operations, and memory overhead.

* Given the focus on SIMD efficiency, why was BGV chosen over CKKS? CKKS is often favored for ML due to its approximate arithmetic and strong SIMD capabilities (e.g., used in EVA [4]). Could the authors elaborate on the rationale for selecting BGV?

* Fairness of comparisons to state-of-the-art methods:
    * The 1023x speedup claim over DSHE on MNIST appears to be an unfair comparison. DSHE uses a deep, complex architecture (akin to ResNet-18) with millions of parameters, whereas HeLutNet uses a very lightweight LUT-based model. Comparing these vastly different model types primarily on latency isn't comparing like-for-like in terms of model complexity or representational power.
     * The Appendix (Table 6) includes DT results for Concrete-ML. It also notes Concrete-ML uses quantization (4-bit/8-bit). Does HeLutNet involve quantization? If comparing to Concrete-ML, clarity on whether quantization-aware training or post-training quantization was used for the Concrete-ML baselines is needed, as this choice significantly impacts final accuracy.
    * In Table 3, HeLutNet (a LUT-based model, conceptually similar to decision trees) is primarily compared against XGBoost models from Concrete-ML. While XGBoost is a strong baseline, I feel it is a bit misleading from a latency perspective to only compare HeLutNet and XGBoost without the Decision Tree (DT) as well. I would suggest combining Table 3 and Table 6 in the main text as this provides a more holistic comparison. Furthermore, are there BGV or CKKS implementations of DTs or similar lightweight models in the literature that could serve as more direct baselines than TFHE-based Concrete-ML?

* A few more performance metrics would be useful to contextualize HeLutNet compared to other methods:
    * What are the memory requirements (e.g., key sizes, ciphertext expansion) for HeLutNet compared to baselines?
    * Could the authors report the total number of homomorphic operations (or perhaps "homomorphic LUT evaluations") performed per inference? This provides a hardware-independent measure of computational cost.
    * Since BGV SIMD operations process multiple data points in parallel, reporting amortized latency per data point (if batching is used/possible) alongside single-instance latency would be informative.

* The paper mentions the challenge of FHE on datasets like CIFAR-10 in the introduction  but does not include CIFAR-10 in its own benchmarks. Given that numerous FHE inference papers (including some cited baselines like EVA) report results on CIFAR-10, its omission is noticeable. Could the authors comment on why CIFAR-10 was excluded and whether HeLutNet was attempted on it?

* Minor point: The statement in the abstract "For the first time, we propose accelerating FHE-based ML by using lookup table (LUT) based ML models" could be slightly refined. While this appears to be the first end-to-end network based on LUTs for FHE, research on using LUTs for evaluating parts of computations or specific functions within FHE does exist [1, 2, 3],. Clarifying the novelty as the first complete network architecture based on LUTs for FHE might be more precise.

Reference: \
[1] Kim, Jaeyun et al. “Privacy-Preserving Embedding via Look-up Table Evaluation with Fully Homomorphic Encryption.” *International Conference on Machine Learning*. 2024 \
[2] Chung et al. “Amortized Large Look-up Table Evaluation with Multivariate Polynomials for Homomorphic Encryption.” *Cryptology ePrint Archive*. 2024 \
[3] Hee Cheon et al. “Tree-based Lookup Table on Batched Encrypted Queries using Homomorphic Encryption.” *Cryptology ePrint Archive*. 2024 \
[4] Dathathri et al. "EVA: An encrypted vector arithmetic language and compiler for efficient homomorphic computation." *Proceedings of the 41st ACM SIGPLAN conference on programming language design and implementation*. 2020

**Questions:**

See weaknesses

---

> ### Author Response · Authors · 2025-11-27
>
> We sincerely thank the reviewer for the insightful comments and helpful suggestions. Our point-to-point responses are as follows.
>
> 1. **HeLutNet's BGV-SIMD vs TFHE-PBS implementation:** We have included a more detailed ablation study comparing HeLutNet's BGV-SIMD approach against a TFHE-PBS implementation of the same LUT network in Appendix A.5 of the revised manuscript.
>
>     **Runtime**
>
>     For a fair performance comparison, we report the BGV runtime using our Microsoft SEAL (C++) followed by Lattigo (Go) implementations.
>
>     |   Dataset   | Model Dimension | Runtime (sec) |            |         |         |         |
>     |:-----------:|:---------------:|:-------------:|:----------:|:-------:|:-------:|:-------:|
>     |             |                 |   Our (BGV)   | Our (TFHE) |         |         |         |
>     |             |                 |     1 Core    |   1 Core   | 16 Core | 32 Core | 48 Core |
>     | MNIST (n=2) |    6000,6000    |  0.008/0.015  |    86.21   |   5.39  |   2.71  |   1.94  |
>     | MNIST (n=3) |    6000,3000    |  0.055/0.126  |    76.60   |   4.63  |   2.34  |   1.87  |
>     |    FMNIST   |    8000,8000    |  0.008/0.015  |   113.84   |   7.17  |   3.60  |   2.59  |
>     |    UCIHAR   |    4200,4200    |  0.008/0.015  |    59.85   |   3.77  |   1.93  |   1.38  |
>     |     QKSD    |    2000,2000    |  0.008/0.015  |    28.94   |   1.80  |   0.92  |   0.64  |
>     |     PTB     |    4000,4000    |  0.008/0.015  |    57.81   |   3.60  |   1.80  |   1.30  |
>
>     **Total homomorphic operation**
>
>     For TFHE-PBS implementation, the number of add and multiply with plaintext operations is (n-1)*N2, and the number of PBS operations is N1+N2, where n is the LUT size and N1 and N2 are the numbers of LUTs in the first and second layers, respectively. For BGV implementation, the number of operations is detailed in response 6.2.
>
>     |   Dataset   | Scheme | No. of Ciphertext | Homomorphic Operations |              |               |          |       |
>     |:-----------:|:------:|:-----------------:|:----------------------:|:------------:|:-------------:|:--------:|:-----:|
>     |             |        |                   |         Add/Sub        | Mult w/plain | Mult w/cipher | Rotation |  PBS  |
>     | MNIST (n=2) |   BGV  |         4         |           16           |       8      |       4       |     0    |  N/A  |
>     |             |  TFHE  |        6000       |          6000          |     6000     |       0       |     0    | 12000 |
>     | MNIST (n=3) |   BGV  |         3         |           20           |       8      |       6       |     2    |  N/A  |
>     |             |  TFHE  |        6000       |          6000          |     6000     |       0       |     0    |  9000 |
>     |     QKSD    |   BGV  |         4         |           16           |       8      |       4       |     0    |  N/A  |
>     |             |  TFHE  |        2000       |          2000          |     2000     |       0       |     0    |  4000 |
>
>     **Memory overhead**
>
>     For a fair comparison, we measured the memory usage of both our C++ and Go BGV implementations. The C++ version, which is based on the Microsoft SEAL library, was profiled using the gperftools and pprof commands. The Go version, built on the Lattigo library, and the TFHE version, implemented with TFHE-GO, were both profiled using Go's standard pprof package.
>
>     |   Dataset   | Scheme |  Total Memory |
>     |:-----------:|:------:|:-------------:|
>     | MNIST (n=2) |   BGV  |  38 MB/51 MB  |
>     |             |  TFHE  |    1102 MB    |
>     | MNIST (n=3) |   BGV  | 196 MB/157MB  |
>     |             |  TFHE  |    1013 MB    |
>     |     QKSD    |   BGV  |  38 MB/51 MB  |
>     |             |  TFHE  |     279 MB    |
>
> 2. **Why was BGV chosen over CKKS?** We choose the BGV scheme because it supports efficient SIMD computation and because floating-point arithmetic is not required in our implementation.

---

> ### Author Response · Authors · 2025-11-27
>
> 3. **Fairness of comparisons to state-of-the-art methods:**
>
>     - **HeLutNet vs DSHE:** We agree that the 1023x speed-up report is unfair and have removed the following sentence from Sec. 5.2. The accuracy and latency of DSHE are retained in Tab. 2 to demonstrate the accuracy vs. latency tradeoff of the existing FHE-based PPML method.
>     - **Does HeLutNet involve quantization?** HeLutNet uses the distributive thermometer encoding for quantizing and binarizing the model inputs. The number of bits used for each dataset is reported in Tab. 8 in Appendix A.2.
>     - **Do the Concrete-ML baselines use quantization-aware training or post-training quantization?** We perform quantization-aware training during Concrete-ML model training.
>     - **Include Decision Tree in Tab. 3:** We have combined information from Tab. 6 (now Tab. 9) into Tab. 3 (now Tab. 4) in the main text. Specifically, we have included the result of the best DT, SVM, and XGBoost models for each dataset directly on Tab. 4.
>     - **Are there BGV or CKKS implementations of DTs or similar lightweight models in the literature?** We have added a comparison with TT-TFHE [1], Homomorphic WiSARDs [2], and Homomorphic Binary Decision Tree (HBDT) [3], including 4 new datasets in the revised manuscript. We hope that these results serve as a more direct baseline and help strengthen our evaluation. In summary, HeLutNet improves upon these lightweight model benchmarks both in terms of accuracy and latency.
>
>         |  Dataset |   Method   | Accuracy (%) | Time (s) |
>         |:--------:|:----------:|:------------:|:--------:|
>         |   MNIST  |  Our (n=2) |     98.38    |    0.008 |
>         |          |   WiSARDs  |     93.76    |    2.184 |
>         | HAM10000 |  Our (n=2) |     74.49    |    0.008 |
>         |          |   WiSARDs  |     69.85    |    6.740 |
>         |   Adult  |  Our (n=2) |     87.10    |    0.008 |
>         |          |   TT-TFHE  |     85.30    |   89.100 |
>         |  Cancer  |  Our (n=2) |     98.25    |    0.008 |
>         |          |   WiSARDs  |     97.30    |    0.026 |
>         |          | HBDT (D=1) |     89.47    |    0.370 |
>         |          |   TT-TFHE  |     97.10    |    1.900 |
>         |          | HBDT (D=4) |     93.64    |    1.650 |
>         | Diabetes |  Our (n=2) |    59.98     |    0.008 |
>         |          |   TT-TFHE  |     57.00    |    72.64 |
>
> 4. **Result on CIFAR-10:** Current SOTA LUT-based model design [4] still can’t achieve high accuracy on larger vision datasets such as CIFAR-10 or ImageNet. For instance, it can only achieve 57.5% on the CIFAR-10 dataset. This limitation of LUT-based networks has been clearly discussed in Sec. 6 of the manuscript. Nevertheless, we believe HeLutNet is the big step toward real-time FHE-based PPML on many real-world small datasets. With concurrent advancement in LUT-based network research, our plan for future work is to scale the HeLutNet approach to future more advanced LUT network design to tackle larger datasets.
>
> 5. **Clarifying the novelty in the abstract:** We have revised the abstract as suggested.
>
> **References**
>
> [1] Benamira, Adrien, et al. "TT-TFHE: a torus fully homomorphic encryption-friendly neural network architecture." arXiv preprint arXiv:2302.01584 (2023).
>
> [2] Neumann, Leonardo, et al. "Homomorphic WiSARDs: Efficient Weightless Neural Network training over encrypted data." International Conference on Applied Cryptography and Network Security. Cham: Springer Nature Switzerland, 2025.
>
> [3] Shin, Hojune, et al. "Fully homomorphic training and inference on binary decision tree and random forest." European Symposium on Research in Computer Security. Cham: Springer Nature Switzerland, 2024
>
> [4] Bacellar, Alan TL, et al. "Differentiable weightless neural networks." arXiv preprint arXiv:2410.11112 (2024).

---

> ### Author Response · Authors · 2025-11-27
>
> 6.	**Performance metrics:**
>
>     - **What are the memory requirements for HeLutNet compared to baselines?** We have detailed the key size, ciphertext size and memory requirement of HeLutNet compared to existing approaches in the following table and in Sec. 5.3 in the revised manuscript.
>
>         |  Dataset |   Method  |  Key Size | Cipher Size | Total Comm. | Total Mem | Accuracy |
>         |:--------:|:---------:|:---------:|:-----------:|:-----------:|:---------:|:--------:|
>         |   MNIST  | Our (n=2) |   1.1 MB  |   1.08 MB   |   2.18 MB   |   38 MB   |  98.38%  |
>         |          |  TT-TFHE  |  766.9 MB |   102.5 MB  |   869.4 MB  |  53.7 MB  |  97.20%  |
>         |          |    SHE    |     ?     |    123 MB   |      ?      |     ?     |  99.54%  |
>         |          |    DiNN   |     ?     |    66 MB    |      ?      |     ?     |  93.71%  |
>         |   Adult  | Our (n=2) |   1.1 MB  |   1.08 MB   |   2.18 MB   |   38 MB   |  87.10%  |
>         |          |  TT-TFHE  |  1.56 GB  |    13 MB    |   1.57 GB   |   15 GB   |  85.30%  |
>         |  Cancer  | Our (n=2) |   1.1 MB  |    1.08MB   |   2.18 MB   |   38 MB   |  98.25%  |
>         |          |  TT-TFHE  | 168.98 MB |    1.7 MB   |  170.68 MB  |     ?     |  97.10%  |
>         | Diabetes | Our (n=2) |   1.1 MB  |    1.08MB   |   2.18 MB   |   38 MB   |  59.98%  |
>         |          |  TT-TFHE  |  1.53 GB  |   37.4 MB   |   1.57 GB   |   6.9 GB  |  57.00%  |
>
>         *? indicates results are not reported.*
>
>     - **Could the authors report the total number of homomorphic operations performed per inference?** We have detailed the number of homomorphic operations as function of the model size in the table below and in Sec. 3.4 in the revised manuscript. As explained in Sec. 3.3, single ciphertext implementation perform less operation but require rotation operation and larger poly_modulus_degree. While the multi-ciphertext implementation perform more operation, the removal of rotation operation and the smaller poly_modulus_degree resulting in large speedup when $n$ is small. From our experiment in Appendix A.4, the multi-cipher implementation is preferred for $n=2$, otherwise the single cipher is preferred.
>
>         | Model Dim | LUT Size | No. of Ciphertext | Homomorphic Operations |              |               |          | Min Poly Modulus Degree |
>         |:---------:|:--------:|:-----------------:|:----------------------:|:------------:|:-------------:|:--------:|:-----------------------:|
>         |           |          |                   |         **Add/Sub**        | **Mult w/plain** | **Mult w/cipher** | **Rotation** |                         |
>         |   $N_1$, $N_2$  |     $n$    |     1 (single)    |       $3(2^n) - 4$       |      $2^n$     |    $2^n - 2$    |    $n-1$   |          $n * N_2$         |
>         |   $N_1, N_2$  |     $n$    |     $n$ (multi)     |       $n(3(2^n)-4)$      |    $n(2^n)$    |   $n(2^n - 2)$  |     $0$    |            $N_2$          |
>
>         The following table compares the number of homomorphic operations needed by HeLutNet vs prior FHE-based PPML work on the MNIST dataset.
>
>         |      Method     | Accuracy | Homomorphic Operations |              |               |          |         |
>         |:---------------:|:--------:|:----------------------:|:------------:|:-------------:|:--------:|:-------:|
>         |                 |          |         **Add/Sub**        | **Mult w/plain** | **Mult w/cipher** | **Rotation** | **Compare** |
>         |    Our (n=2)    |  98.38%  |           16           |       8      |       4       |     0    |    0    |
>         | Orion (MLP) [6] |  98.03%  |            ?           |       ?      |       ?       |    70    |    0    |
>         |     SHE [7]     |  99.54%  |           19K          |      945     |       0       |    19K   |    3K   |
>         |     DiNN [7]    |  93.71%  |           8K           |      8K      |       40      |     0    |    0    |
>         |     Lola [7]    |  98.95%  |          12.5K         |     10.5K    |      1.6K     |     0    |    0    |
>
>     - **Since BGV SIMD operations process multiple data points in parallel, reporting amortized latency per data point (if batching is used/possible):** HeLutNet only utilizes SIMD to speed up inference of a single input sample. Batching is currently not used in our current design.
>
> **References**
>
> [6] Ebel, Austin, Karthik Garimella, and Brandon Reagen. "Orion: A Fully Homomorphic Encryption Framework for Deep Learning." Proceedings of the 30th ACM International Conference on Architectural Support for Programming Languages and Operating Systems, Volume 2. 2025
>
> [7] Lou, Qian, and Lei Jiang. "She: A fast and accurate deep neural network for encrypted data." Advances in neural information processing systems 32 (2019).

---

### Official Review · Reviewer_a1MN · 2025-10-31

**Soundness:** 3
**Presentation:** 2
**Contribution:** 3
**Rating:** 6
**Confidence:** 3

**Summary:**

The paper proposes to HELUTNET framework for decreasing the latency of FHE-based PPML by adapting Look Up Table-based ML models to FHE. This approach, based on the BGV scheme, converts LUT ML models to low-latency FHE programs while utilizing SIMD operations in BGV. The paper evaluates the proposed approach against various baselines w.r.t latency and accuracy on lightweight vision, speech, and tabular datasets.

**Strengths:**

1)The paper presents a practical approach for FHE-based privacy-preserving applications on lightweight datasets, showing good improvement over the baselines in terms of the latency on the MNIST and Fashion MNIST datasets.
2)The approach is validated on a wide variety of lightweight datasets, showing its ability to generalize across modalities.
3)The latency problems of FHE are well motivated, and the proposed approach and FHE concepts are clearly presented.

**Weaknesses:**

1) The related work section is not thorough; the context of the prior work on improving the latency of FHE-based PPML is not established.
2)Prior work has used LUTs in an FHE setting for PPML; they have not been discussed or cited. Some examples are given below:
	a) Privacy-Preserving Embedding via Look-up Table Evaluation with Fully Homomorphic Encryption
	b) Homomorphic WiSARDs: Efficient Weightless Neural Network training over encrypted data
	c) TT-TFHE: a Torus Fully Homomorphic Encryption-Friendly Neural Network Architecture
3)The paper claims it has better latency than the SOTA MPC protocols without any comparison results presented in the paper.
4)The limitations of expressivity of LUT-based ML models could prohibit scaling to larger datasets.

**Questions:**

For the datasets other than MNIST and fashion MNIST, the approach is compared only against  Concrete-ML. Is it the SOTA FHE-based PPML method for these datasets?. If not, what were the reasons for not using other methods as baselines for these datasets?

---

> ### Comment · Reviewer_a1MN · 2025-11-25
> **No response received**
>
> Since the authors have not provided any comments or answers, the review rating is unchanged.

---

> > ### Author Response · Authors · 2025-11-25
> >
> > We apologize for the delay in providing a response and sincerely appreciate the effort the reviewer has put into reviewing our work. We have conducted many new experiments based on all reviewers' feedback and will provide responses to all reviewers today.

---

> ### Author Response · Authors · 2025-11-27
>
> We thank the reviewer for the effort put into reviewing our work and for pointing out several related works for comparison with our methods. We provide point-to-point responses and detail our modification to the manuscript below.
>
> 1. **Comparison with TT-TFHE, Homomorphic WiSARDs:** Thank you for pointing out these related works. We have included new experiment results comparing with TT-TFHE [1], Homomorphic WiSARDs [2], and Homomorphic Binary Decision Tree (HBDT) [3] in the table below and in Tab. 3 in the revised manuscript. We hope that these new results will help strengthen our evaluation and further demonstrate HeLutNet's applicability to diverse application scenarios.
>
> 	|  Dataset |   Method   | Accuracy (%) | Time (s) |
> 	|:--------:|:----------:|:------------:|:--------:|
> 	|   MNIST  |  Our (n=2) |     98.38    |    0.008 |
> 	|          |   WiSARDs  |     93.76    |    2.184 |
> 	| HAM10000 |  Our (n=2) |     74.49    |    0.008 |
> 	|          |   WiSARDs  |     69.85    |    6.740 |
> 	|   Adult  |  Our (n=2) |     87.10    |    0.008 |
> 	|          |   TT-TFHE  |     85.30    |   89.100 |
> 	|  Cancer  |  Our (n=2) |     98.25    |    0.008 |
> 	|          |   WiSARDs  |     97.30    |    0.026 |
> 	|          | HBDT (D=1) |     89.47    |    0.370 |
> 	|          |   TT-TFHE  |     97.10    |    1.900 |
> 	|          | HBDT (D=4) |     93.64    |    1.650 |
> 	| Diabetes |  Our (n=2) |    59.98     |    0.008 |
> 	|          |   TT-TFHE  |     57.00    |    72.64 |
>
> 2. **Comparison to SOTA MPC works:** We present a comparison in terms of latency and communication cost to recent SOTA MPC works in the following table. For all MPC works, we choose the latency in the LAN setting for a fair comparison. The results demonstrate that HeLutNet achieves both a lower communication payload size and lower latency compared to the SOTA MPC methods.
>
> 	| Method      | Model     | Accuracy (%) | (LAN) Time (ms) | Communication (MB) |
> 	|-------------|-----------|----------|-----------------|--------------------|
> 	| Our         | DWN (n=2) |   98.38% |               8 |               2.18 |
> 	| SWIFT [4]   | CNN       |        ? |           24.14 |                  ? |
> 	| Cheetah [5] | CNN       |        ? |           42.88 |                  ? |
> 	| MD-ML [6]   | LeNet     |    98.43 |              80 |               9.46 |
> 	| SPDZ [7]    | LeNet     |        ? |             274 |              18.99 |
> 	*? indicates results are not reported.*
>
> 3. **Is Concrete-ML the SOTA FHE-based PPML method for these datasets? If not, why using Concrete-ML?** There isn’t any established baseline on these datasets in the FHE-based inference scenario. Concrete-ML is chosen as it is the current SOTA open-source implementation of FHE-based privacy-preserving DT, SVM, and XGB models. For all datasets, we train DT, SVM, and XGB models to match the best plaintext accuracy of these datasets reported in [8, 9] and report privacy-preserving inference time based on Concrete-ML. For better clarity, we have revised Tab. 3 (now Tab. 4 in the updated manuscript) to show accuracy and runtime of all models, incorporating results from Tab. 6 (now Tab. 9) in Appendix A.3 into the main text.
>
> **References**
>
> [1] Benamira, Adrien, et al. "TT-TFHE: a torus fully homomorphic encryption-friendly neural network architecture." arXiv preprint arXiv:2302.01584 (2023).
>
> [2] Neumann, Leonardo, et al. "Homomorphic WiSARDs: Efficient Weightless Neural Network training over encrypted data." International Conference on Applied Cryptography and Network Security. Cham: Springer Nature Switzerland, 2025.
>
> [3] Shin, Hojune, et al. "Fully homomorphic training and inference on binary decision tree and random forest." European Symposium on Research in Computer Security. Cham: Springer Nature Switzerland, 2024.
>
> [4] Fu, Yu, et al. "Swift: Fast Secure Neural Network Inference With Fully Homomorphic Encryption." IEEE Transactions on Information Forensics and Security (2025).
>
> [5] Huang, Zhicong, et al. "Cheetah: Lean and fast secure {Two-Party} deep neural network inference." 31st USENIX Security Symposium (USENIX Security 22). 2022.
>
> [6] Yuan, Boshi, et al. "{MD-ML}: Super Fast {Privacy-Preserving} Machine Learning for Malicious Security with a Dishonest Majority." 33rd USENIX Security Symposium (USENIX Security 24). 2024.
>
> [7] Damgård, Ivan, et al. "New primitives for actively-secure MPC over rings with applications to private machine learning." 2019 IEEE Symposium on Security and Privacy (SP). IEEE, 2019.
>
> [8] Bacellar, Alan TL, et al. "Differentiable weightless neural networks." Proceedings of the 41st International Conference on Machine Learning. 2024.
>
> [9] Narkthong, Nuntipat, et al. "Microvsa: An ultra-lightweight vector symbolic architecture-based classifier library for always-on inference on tiny microcontrollers." Proceedings of the 29th ACM International Conference on Architectural Support for Programming Languages and Operating Systems, Volume 2. 2024.

---

### Official Review · Reviewer_ikbp · 2025-11-02

**Soundness:** 2
**Presentation:** 3
**Contribution:** 1
**Rating:** 2
**Confidence:** 4

**Summary:**

The paper presents an implementation of LUT-based FHE for PPML inference, which is technically sound but suffers from significant limitations in novelty and practical utility. While the authors claim to be the first to propose LUT-based models for FHE-accelerated inference, lookup table operations have been a standard technique in TFHE/FHEW schemes (of course, this paper adopted BGV, instead of TFHE), making the technical contribution somewhat incremental and less groundbreaking than presented. The claimed speedup on MNIST is also questionable when amortized fairly against recent PPML baselines, and without adequately accounting for model size differences, the reported 17.7× improvement appears to be primarily an artifact of choosing a straightforward network architecture. Most critically, the complete absence of evaluation on CIFAR-10—which is arguably still a modest dataset in today's PPML research—reveals a fundamental scalability barrier that the authors sidestep rather than address, undermining claims of practical applicability. The core trade-off of using shallow 2-layer models severely sacrifices representational capacity, rendering complex real-world inference tasks infeasible, yet this is presented as a feature rather than honestly discussed as a severe limitation. Compared to concurrent work and other recent FHE frameworks that handle deeper networks and larger datasets, this paper addresses a problem that feels increasingly dated and somewhat orthogonal to the field's current direction. The experimental scope is too narrow, the model architectures are too simplistic, and the practical impact is too limited to warrant acceptance at a top-tier venue, as the work essentially demonstrates that LUTs can be fast in FHE at the cost of the expressiveness needed for non-trivial applications.

**Strengths:**

The paper presents a technical solution for efficiently implementing LUTs in FHE using SIMD operations within the BGV scheme, enabling faster inference on MNIST without requiring expensive bootstrapping.

**Weaknesses:**

The paper's evaluation is severely limited by its inability to scale beyond MNIST, with no results on CIFAR-10 and an acknowledged impracticality for larger datasets, thereby fundamentally undermining claims of real-world applicability. The core trade-off of using ultra-shallow 2-layer networks to achieve speed is sacrificing the representational power needed for non-trivial tasks, and the amortized performance gains over recent SOTA baselines become questionable once you account for the vast differences in model complexity and input dimensionality.

**Questions:**

Does the paper provide any results on scaling this approach to larger models and datasets? What are the core problems and limitations that arise when trying to do so?

---

> ### Author Response · Authors · 2025-11-26
>
> We appreciate the reviewer's constructive comments and different point of view on our work and have provided a point-by-point response below.
>
> > “lookup table operations have been a standard technique in TFHE/FHEW schemes (of course, this paper adopted BGV, instead of TFHE), making the technical contribution somewhat incremental and less groundbreaking than presented.”
>
> While LUT is a standard technique in TFHE, there is still limited work on how to use it to implement privacy-preserving inference efficiency. More importantly, we are first to demonstrate in this paper that direct implementation of a LUT-based network using TFHE is extremely slow (see the table below, which is also included as Appendix A.5 in the revised manuscript) and propose a much more efficient method based on the BGV/BFV scheme.
>
> |   Dataset   | Model Dimension | Runtime (sec) |            |         |         |         |
> |:-----------:|:---------------:|:-------------:|:----------:|:-------:|:-------:|:-------:|
> |             |                 |   Our (BGV)   | Our (TFHE) |         |         |         |
> |             |                 |     1 Core    |   1 Core   | 16 Core | 32 Core | 48 Core |
> | MNIST (n=2) |    6000,6000    |     0.008     |    86.21   |   5.39  |   2.71  |   1.94  |
> | MNIST (n=3) |    6000,3000    |     0.055     |    76.60   |   4.63  |   2.34  |   1.87  |
> |    FMNIST   |    8000,8000    |     0.008     |   113.84   |   7.17  |   3.60  |   2.59  |
> |    UCIHAR   |    4200,4200    |     0.008     |    59.85   |   3.77  |   1.93  |   1.38  |
> |     QKSD    |    2000,2000    |     0.008     |    28.94   |   1.80  |   0.92  |   0.64  |
> |     PTB     |    4000,4000    |     0.008     |    57.81   |   3.60  |   1.80  |   1.30  |
>
> Moreover, the evaluation results show that HeLutNet outperforms all other neural networks and weightless networks implemented using the TFHE scheme.
>
> |  Dataset |  Method | FHE Scheme / Library  | Accuracy (%) | Time (s) |
> |:--------:|:-------:|:---------------------:|:------------:|:--------:|
> |   MNIST  |   Our   |       BGV (SEAL)      |     98.38    |    0.008 |
> |          |   SHE   |      TFHE (TFHE)      |     93.76    |    2.184 |
> | HAM10000 |   Our   |       BGV (SEAL)      |     74.49    |    0.008 |
> |          | WiSARDs |     TFHE (MOSFHET)    |     69.85    |    6.740 |
> |   Adult  |   Our   |       BGV (SEAL)      |     87.10    |    0.008 |
> |          | TT-TFHE |    TFHE (Concrete)    |     85.30    |   89.100 |
> |  Cancer  |   Our   |       BGV (SEAL)      |     98.25    |    0.008 |
> |          | TT-TFHE |    TFHE (Concrete)    |     97.30    |    0.026 |
> |          | WiSARDs |     TFHE (MOSFHET)    |     97.10    |    1.900 |
> | Diabetes |   Our   |       BGV (SEAL)      |     59.98    |    0.008 |
> |          | TT-TFHE |    TFHE (Concrete)    |     57.00    |    72.64 |
>
> > “The claimed speedup on MNIST is also questionable when amortized fairly against recent PPML baselines, and without adequately accounting for model size differences, the reported 17.7× improvement appears to be primarily an artifact of choosing a straightforward network architecture.”
>
> Even though our method is simpler, the reported speedup in terms of runtime is fair because 1) the evaluation is performed on the same dataset and a train/test split and 2) our model achieves comparable accuracy to the compared baseline.
>
> > “Most critically, the complete absence of evaluation on CIFAR-10—which is arguably still a modest dataset in today's PPML research—reveals a fundamental scalability barrier that the authors sidestep rather than address, undermining claims of practical applicability. The core trade-off of using shallow 2-layer models severely sacrifices representational capacity, rendering complex real-world inference tasks infeasible, yet this is presented as a feature rather than honestly discussed as a severe limitation.”
>
> Our goal is not to demonstrate an FHE-based PPML system supporting larger or more complex datasets nor to increase the accuracy of FHE-based PPML inference on common datasets such as MNIST, CIFAR-10, or ImageNet. Rather, we aim to address the significant runtime overhead of the FHE-based PPML method, which usually takes seconds to minutes to complete. For instance, to the best of our knowledge, the fastest runtime report on CPU of any FHE-based PPML inference method on the MNIST dataset that can reach 98-99% accuracy is Orion [1] at 0.23 seconds, which is still much slower compared to plain-text inference.
>
> We also would like to argue that the current LUT-based model, while being shallow and less complex, can achieve SOTA accuracy on many small real-world datasets. For instance, in this work we demonstrate competitive accuracy on multiple vision datasets (MNIST, Fashion-MNIST, HAM10000), speech datasets (QKSD, FSDD), health datasets (PTB, UCIHAR, Cancer, Diabetes), and several tabular datasets.

---

> ### Author Response · Authors · 2025-11-26
>
> > “Compared to concurrent work and other recent FHE frameworks that handle deeper networks and larger datasets, this paper addresses a problem that feels increasingly dated and somewhat orthogonal to the field's current direction.”
>
> We respectfully disagree with this proposition. There are already many works on extending FHE-based PPML to more complex datasets like ImageNet or on reducing its runtime from hours to minutes or seconds. However, there isn’t any work yet to achieve a real-time single-digit millisecond-level runtime per inference on any small or large dataset. This is the long-unsolved research gap we are trying to address.
>
> > “The experimental scope is too narrow, the model architectures are too simplistic, and the practical impact is too limited to warrant acceptance at a top-tier venue, as the work essentially demonstrates that LUTs can be fast in FHE at the cost of the expressiveness needed for non-trivial applications.”
>
> We argue that showing a LUT-based model can be fast in FHE and achieve high accuracy on many small real-world datasets is a significant contribution, as no prior work has presented a practical FHE-based PPML inference framework that can run in real-time (single-digit milliseconds) on either small or large datasets (see the table below for a runtime comparison to the current fastest work on each dataset). Our paper demonstrates new, relevant, impactful knowledge as well as achieves state-of-the-art performance.
>
> |  Dataset |   Method  | Accuracy (%) | Δ Accuracy | Time (s) |  Speedup  |
> |:--------:|:---------:|:------------:|:----------:|:--------:|:---------:|
> |   MNIST  | Our (n=2) |     98.38    |    -0.24   |    0.008 |    28.75x |
> |          | Our (n=3) |     98.50    |    -0.12   |    0.055 |     4.18x |
> |          |   Orion   |     98.62    |            |    0.230 |           |
> | HAM10000 | Our (n=2) |     74.49    |    +4.64   |    0.008 |    842.5x |
> |          |  WiSARDs  |     69.85    |            |    6.740 |           |
> |   Adult  | Our (n=2) |     87.10    |    +1.80   |    0.008 | 11,137.5x |
> |          |  TT-TFHE  |     85.30    |            |   89.100 |           |
> |  Cancer  | Our (n=2) |     98.25    |    +0.95   |    0.008 |     3.25x |
> |          |  WiSARDs  |     97.30    |            |    0.026 |           |
> | Diabetes | Our (n=2) |     59.98    |    +2.98   |    0.008 |    9,080x |
> |          |  TT-TFHE  |     57.00    |            |   72.640 |           |
>
> > “Does the paper provide any results on scaling this approach to larger models and datasets? What are the core problems and limitations that arise when trying to do so?”
>
> Current SOTA LUT-based model designs (e.g., [2] and [3]) still can’t achieve high accuracy on larger vision datasets such as CIFAR-10 or ImageNet. For instance, [2] can only achieve 57.5%, and [3] can only achieve 62.14% on the CIFAR-10 dataset. This limitation of LUT-based networks has been clearly discussed in Sec. 6 of the manuscript. Nevertheless, we believe HeLutNet is the first big step toward real-time FHE-based PPML on many real-world small datasets. With concurrent advancement in LUT-based network research, our plan for future work is to scale the HeLutNet approach to tackle larger datasets.
>
> We hope that these responses help clarify the goal and main contributions of our work. Kindly let us know if you would like us to make any other clarification or changes to the manuscript.
>
> **References**
>
> [1] Austin Ebel, Karthik Garimella, and Brandon Reagen. 2025. Orion: A Fully Homomorphic Encryption Framework for Deep Learning. In Proceedings of the 30th ACM International Conference on Architectural Support for Programming Languages and Operating Systems, Volume 2 (ASPLOS '25). Association for Computing Machinery, New York, NY, USA, 734–749. https://doi.org/10.1145/3676641.3716008
>
> [2] Alan T. L. Bacellar, Zachary Susskind, Mauricio Breternitz Jr., Eugene John, Lizy K. John, Priscila M. V. Lima, and Felipe M. G. França. 2024. Differentiable weightless neural networks. In Proceedings of the 41st International Conference on Machine Learning (ICML'24), Vol. 235. JMLR.org, Article 90, 2277–2295.
>
> [3] Felix Petersen, Christian Borgelt, Hilde Kuehne, and Oliver Deussen. 2022. Deep differentiable logic gate networks. In Proceedings of the 36th International Conference on Neural Information Processing Systems (NIPS '22). Curran Associates Inc., Red Hook, NY, USA, Article 146, 2006–2018.

---

> > ### Comment · Reviewer_ikbp · 2025-11-27
> > **I will keep my original evaluation**
> >
> > I’m still not convinced by the very strong claim that TFHE-based LUT implementations are inherently much slower than BGV/CKKS---TFHE is natively built around LUT-style bootstrapping, there is no evidence that your TFHE baseline uses a state-of-the-art fast TFHE library or carefully tuned parameters, and the work is not clearly positioned against the rapidly growing line of functional-bootstrapping/LUT results for BGV/CKKS, where recent CKKS-based LUT schemes already report sub-millisecond amortized latency per input (e.g., https://eprint.iacr.org/2024/1623). On top of that, it’s not clear whether your reported latencies (e.g., 13 ms) are amortized per sample over a large batch or actual single-input wall-clock times, so the measurement methodology and the basis for comparison with prior work should really be clarified.
> >
> > In addition, my question remains: what is the real practical meaning of getting smaller latency on MNIST by switching to a much simpler model instead of using the existing FHE-friendly models, especially given that this “small model” strategy is unlikely to carry over to CIFAR-10 or more realistic tasks? The manuscript and the rebuttal do not really discuss the FHE-level scaling bottlenecks (LUT input size, LUT table size, ciphertext slots, rotations, depth, parameters/key sizes, memory, etc.), nor do they provide a qualitative scaling analysis. Taken together, the limitations of the chosen approach feel quite clear to me, so I still see the contribution as fairly minor and will keep my original evaluation.

---

> ### Author Response · Authors · 2025-12-04
>
> Thank you for the prompt response. We have incorporated new empirical results and detailed explanations into the revised manuscript to address all concerns, clarify potential misunderstandings, and respectfully substantiate our claims where we hold a differing view.
>
> > “I’m still not convinced by the very strong claim that TFHE-based LUT implementations are inherently much slower than BGV/CKKS---TFHE is natively built around LUT-style bootstrapping, there is no evidence that your TFHE baseline uses a state-of-the-art fast TFHE library or carefully tuned parameters”
>
> We never claim that TFHE-based LUT implementations are always slower than BGV/CKKS. Indeed, Sec. 3.1 explains the trade-off and the reason why we believe the BGV is more suitable for our use case. To support this, Tab. 3 confirms that the HeLutNet BGV-based implementation outperforms all prior FHE-based PPML works based on the BGV, CKSS, and TFHE schemes, and Appendix A.5 shows that runtime of our BGV-based implementation is much faster than the TFHE-based implementation.
>
> HeLutNet's TFHE-based implementation adopted the TFHE-go library, which, according to our benchmark, performs the programmable bootstrapping faster than other SOTA libraries, including TFHE (C++), TFHE-rs, and OpenFHE. As the main contribution of this work is not to compare FHE schemes, we didn’t claim and provide evidence to suggest that our TFHE-based implementation is the fastest or uses the best-tuned parameters.
>
>  > “the work is not clearly positioned against the rapidly growing line of functional-bootstrapping/LUT results for BGV/CKKS, where recent CKKS-based LUT schemes already report sub-millisecond amortized latency per input (e.g., https://eprint.iacr.org/2024/1623).”
>
> We believe the mentioned paper (CKKS-based LUT schemes) is unrelated to our work, as they focused on implementing independent LUT without any discussion on how to use it to perform PPML inference nor how to efficiently implement a LUT network (HeLutNet’s packing and reordering strategy).
>
> > “On top of that, it’s not clear whether your reported latencies (e.g., 13 ms) are amortized per sample over a large batch or actual single-input wall-clock times, so the measurement methodology and the basis for comparison with prior work should really be clarified.”
>
> All latencies reported in the paper are actual single-input wall-clock times without batching. We have updated all related table captions to clarify this point.
>
> > “What is the real practical meaning of getting smaller latency on MNIST by switching to a much simpler model instead of using the existing FHE-friendly models, especially given that this “small model” strategy is unlikely to carry over to CIFAR-10 or more realistic tasks?”
>
> This paper is not about getting lower latency on MNIST (which we indeed perform inference on faster than all prior works). Rather, we proposed a new FHE-friendly algorithm and demonstrated that it enables, for the first time, real-time FHE-based PPML inference on 20 datasets across 4 categories (vision, speech, health, and tabular). For instance, HeLutNet is the first framework to achieve real-time FHE-based privacy-preserving speech/wake word recognition (8.3 ms inference time per sample on a single-core CPU without any batching tested on the Qualcomm Speech Keyword Dataset and Free-Spoken Digit Dataset).
>
> It is well known that latency is a key factor limiting the practicality of FHE-based PPML inference. While supporting large datasets like CIFAR-10 and ImageNet while maintaining real-time performance is the ideal goal, this is currently unachievable by any prior works. We believe HeLutNet represents a significant step forward for small-to-medium datasets, and this scaling limitation is documented in Section 6.
>
> > “The manuscript and the rebuttal do not really discuss the FHE-level scaling bottlenecks (LUT input size, LUT table size, ciphertext slots, rotations, depth, parameters/key sizes, memory, etc.), nor do they provide a qualitative scaling analysis.”
>
> The total number of homomorphic operations (add, multiply, rotation), multiplicative depth, and number of ciphertext slots for different model sizes are reported in Tab. 2 in Sec. 3.4. Key size and memory usage are reported in Sec. 5.3 (Tab. 5-6). Runtimes for different model sizes are provided in Appendix A.4 in the updated manuscript.

---

### Official Review · Reviewer_tHCh · 2025-11-08

**Soundness:** 2
**Presentation:** 3
**Contribution:** 2
**Rating:** 4
**Confidence:** 4

**Summary:**

To build a privacy-preserving AI model using homomorphic encryption, one must handle numerous non-linear operations, which significantly increase computational overhead. Consequently, homomorphic encryption–based models typically require longer inference times. In this paper, the authors propose a model that leverages lookup tables (LUTs) to enable efficient computation under the BGV scheme.
As a result, the proposed method achieves tens of times faster performance compared to previous approaches, while maintaining competitive accuracy on lightweight datasets such as MNIST.

**Strengths:**

First of all, I believe the DWN, or more precisely, the efficient construction of the LUT, is a meaningful contribution. There have not been many studies that implemented LUTs efficiently. Typically, LUT-based methods rely on interpolation polynomials or quantization followed by comparison-based approaches. In contrast, this paper presents an optimization specifically tailored to the DWN structure, which I consider to be a clear and notable contribution.

**Weaknesses:**

The implementation of LUT branching in this paper is rather simple. Such a design heavily depends on the packing scheme, which in turn affects the communication cost. Since the encrypted data is transmitted after thermometer encoding, the overall communication overhead is inevitably large. Although it may still be lower than that of MPC-based approaches, the paper does not provide any discussion or quantitative analysis comparing the communication cost with other FHE-based methods.

Moreover, this paper is not the first to implement WNNs under homomorphic encryption, which further weakens the novelty. Another potential issue is that, due to the dependency on the packing scheme, the model structure could be partially inferred from the packing pattern itself. While the reordering strategy to reduce the number of rotations is indeed a strong contribution, it could also expose relative positional information that may leak insights into the network’s architecture.

Finally, the most critical limitation is the simplicity of the dataset used. The proposed method’s computational cost will likely explode as the model architecture or dataset complexity increases, making it difficult to argue that the approach is efficient based on experiments limited to MNIST. Since this is not the first work of its kind, the paper should at least demonstrate applicability on a more complex dataset or architecture—for example, CIFAR-10—and evaluate the method with a larger LUT size (greater than 3) to establish its scalability and effectiveness.

**Questions:**

I believe a comparison with other FHE studies implementing LUT or comparison operations is necessary. From my perspective, Decision Tree (DT) models share strong similarities with the proposed approach, as both derive outputs based on predefined, learned rules.
In particular, the branching mechanism in DTs appears conceptually very similar to that of this paper (e.g., Level-up style branching). Therefore, I would like to ask whether the proposed method could be extended or adapted to Decision Tree models, and if so, what modifications would be required to make such an extension feasible.

---

> ### Author Response · Authors · 2025-11-26
>
> We sincerely thank the reviewer for the insightful comments and suggestions regarding additional experiments on communication cost. Our point-to-point responses are as follows.
>
> 1. **Communication of HeLutNet vs other FHE-based PPML methods:** We detail all key sizes and all ciphertext sizes, which dictates the communication overhead of HeLutNet for some representative model configurations in the table below and Sec. 5.3 of the revised manuscript. Note that thermometer encoding doesn't affect the key and ciphertext sizes, as they solely depend on the model dimension (number of LUTs in each layer and LUT size), i.e., only the bit that is used as an input to LUT needs to be packed and encrypted. The result shows that HeLutNet communication cost is the lowest compared to all existing methods.
>
>     |  Dataset |   Method  |  Key Size | Ciphertext Size | Total Comm. | Accuracy |
>     |:--------:|:---------:|:---------:|:-----------:|:-----------:|:--------:|
>     |   MNIST  | Our (n=2) |   1.1 MB  |    1.08MB   |   2.18 MB   |  98.38%  |
>     |          |  SHE [1]  |     ?     |    123 MB   |      ?      |  99.54%  |
>     |          |  DiNN [1] |     ?     |    66 MB    |      ?      |  93.71%  |
>     |   Adult  | Our (n=2) |   1.1 MB  |    1.08MB   |   2.18 MB   |  87.10%  |
>     |          |  TT-TFHE  |  1.56 GB  |    13 MB    |   1.57 GB   |  85.30%  |
>     |  Cancer  | Our (n=2) |   1.1 MB  |    1.08MB   |   2.18 MB   |  98.25%  |
>     |          |  TT-TFHE  | 168.98 MB |    1.7 MB   |  170.68 MB  |  97.10%  |
>     | Diabetes | Our (n=2) |   1.1 MB  |    1.08MB   |   2.18 MB   |  59.98%  |
>     |          |  TT-TFHE  |  1.53 GB  |   37.4 MB   |    1.57GB   |  57.00%  |
>
>
> 2. **Comparison with other FHE studies implementing LUT and DT:** We have included new results to compare with previous LUT-based work mentioned by reviewer a1MN (TT-TFHE [2], Homomorphic WiSARDs [3]) and Homomorphic Binary Decision Tree (HBDT) [4] in the table below and in Tab. 2 in the revised manuscript.  These results show that HeLutNet achieves superior accuracy while significantly reducing runtime on all datasets.
>
>     |  Dataset |   Method   | Accuracy (%) | Time (s) |
>     |:--------:|:----------:|:------------:|:--------:|
>     |   MNIST  |  Our (n=2) |     98.38    |    0.008 |
>     |          |   WiSARDs  |     93.76    |    2.184 |
>     | HAM10000 |  Our (n=2) |     74.49    |    0.008 |
>     |          |   WiSARDs  |     69.85    |    6.740 |
>     |   Adult  |  Our (n=2) |     87.10    |    0.008 |
>     |          |   TT-TFHE  |     85.30    |    89.10 |
>     |  Cancer  |  Our (n=2) |     98.25    |    0.008 |
>     |          |   WiSARDs  |     97.30    |    0.026 |
>     |          | HBDT (D=1) |     89.47    |    0.370 |
>     |          |   TT-TFHE  |     97.10    |    1.900 |
>     |          | HBDT (D=4) |     93.64    |    1.650 |
>     | Diabetes |  Our (n=2) |    59.98     |    0.008 |
>     |          |   TT-TFHE  |     57.00    |    72.64 |
>
>     Based on the comments from all reviewers and to the best of our knowledge, these are the three additional works that we should cite. Kindly let us know if the reviewer would like us to add a comparison with any other work.
>
> 3. **Will HeLutNet’s packing and reorder strategy leak model structure/network architecture?:** HeLutNet’s goal is to protect the input. It does not protect the model architecture and weight, which is stored and known by another party (e.g., server). This scenario aligns with prior FHE-based PPML approaches.
> Nevertheless, if the ciphertext (encrypted after our packing and reorder strategy) were to be captured by an attacker during transmission to the server, it will only reveal the number of inputs of the LUT but not the number of LUTs in each layer and the LUT input/output mapping (model weight). The reason is that the size (poly_modulus_degree) and the number of ciphertexts transmitted are identical for all models with the same LUT size (number of inputs of the LUT). Specifically, our ciphertext size depends on the required computation depth, which depends on the level of the mux tree (number of inputs of the LUT). The number of ciphertexts transmitted also depends on the number of inputs of the LUT (a model with n=3 needs to transmit 3 ciphertexts after packing and reordering).
> 4. **Evaluation only on small dataset. Efficiency may be due to simplicity of the dataset:** Our evaluation dataset isn’t limited to MNIST, as we have also included other vision datasets (Fashion-MNIST), speech datasets (QKSD, FSDD), and health datasets (PTB, UCIHAR) in Tab. 5 and Appendix A.2 in the original manuscript. To strengthen the evaluation, we have included a comparison with TT-TFHE, homomorphic WiSARDs, and HBDT, including 4 new datasets. We hope that these new results help demonstrate HeLutNet's applicability to diverse application scenarios.

---

> ### Author Response · Authors · 2025-11-26
>
> 5. **Does HeLutNet work with larger LUT size (>3)?** Yes, HeLutNet works with any LUT size. However, our experiments showed that training the model with an LUT larger than 3 doesn’t yield much improvement in terms of accuracy while significantly increasing the inference time. Therefore, it is better to use a larger number of small LUTs (which doesn’t incur a runtime penalty due to our ciphertext packing strategy) rather than a smaller number of larger LUTs.
> The following table demonstrates the accuracy and runtime (MS SEAL-based implementation using 1 thread) on the MNIST dataset using different LUT sizes from 2 to 6.
>
>     **MNIST**
>
>     | n (LUT Size) | Architecture | Accuracy (%) | Runtime (ms) |
>     |:------------:|:------------:|:------------:|:------------:|
>     |       2      |   6000,6000  |    98.38%    |     8.32     |
>     |       3      |   6000,3000  |    98.50%    |     55.04    |
>     |       4      |   5000,2500  |    98.38%    |    161.68    |
>     |       5      |   4000,2000  |    98.43%    |    1129.33   |
>     |       6      |   2000,1000  |    98.47%    |    2093.95   |
>
>     **Fashion-MNIST**
>
>     | n (LUT Size) | Architecture | Accuracy (%) | Runtime (ms) |
>     |:------------:|:------------:|:------------:|:------------:|
>     |       2      |   8000,8000  |    89.75%    |     8.32     |
>     |       3      |   5000,2500  |    90.03%    |     55.04    |
>     |       4      |   4000,2000  |    89.70%    |    161.68    |
>     |       5      |   3000,1500  |    89.97%    |    1129.33   |
>     |       6      |   2000,1000  |    89.52%    |    2093.95   |
>
> 6. **LUT-based model vs decision tree:** While our realization of LUT with a multiplexor tree makes it seem like a decision tree, LUT is fundamentally different from a decision tree. Decision tree models derive output based on predefined learned rules. It determines the next rule by creating a branch (e.g., using a comparison operation) as a main building block, which is complex to implement in encrypted fashion. On the other hand, the LUT-based model learns the input/output mapping for each LUT. LUT can be implemented as a multiplexor tree with a fixed structure. Each multiplexer generates output to be used as an input of the next multiplexer without any branching. While an LUT network can be converted to a large decision tree, the decision tree cannot be directly converted to an LUT.
>
> **References**
>
> [1] Qian Lou and Lei Jiang. 2019. SHE: a fast and accurate deep neural network for encrypted data. Proceedings of the 33rd International Conference on Neural Information Processing Systems. Curran Associates Inc., Red Hook, NY, USA, Article 900, 10035–10043.
>
> [2] Benamira, Adrien, Tristan Guérand, Thomas Peyrin, and Sayandeep Saha. "TT-TFHE: a torus fully homomorphic encryption-friendly neural network architecture." arXiv preprint arXiv:2302.01584 (2023).
>
> [3] Neumann, Leonardo, Antonio Guimarães, Diego F. Aranha, and Edson Borin. "Homomorphic WiSARDs: Efficient Weightless Neural Network training over encrypted data." In International Conference on Applied Cryptography and Network Security, pp. 309-338. Cham: Springer Nature Switzerland, 2025.
>
> [4] Hojune Shin, Jina Choi, Dain Lee, Kyoungok Kim, and Younho Lee. 2024. Fully Homomorphic Training and Inference on Binary Decision Tree and Random Forest. In Computer Security – ESORICS 2024: 29th European Symposium on Research in Computer Security, Bydgoszcz, Poland, September 16–20, 2024, Proceedings, Part III. Springer-Verlag, Berlin, Heidelberg, 217–237. https://doi.org/10.1007/978-3-031-70896-1_11

---

### Author Response · Authors · 2025-12-01

We sincerely thank all reviewers for the effort put into reviewing our work. In addition to our response to the individual reviewer, we would like to summarize several key improvements and modifications made to the manuscript, as detailed below.

1. **Expanded SOTA Comparisons:** We have added comparisons with recent SOTA FHE weightless neural networks and decision trees as requested by several reviewers, including WiSARDs (ACNS 2025), TT-TFHE (accepted to TMLR on Nov 28, 2025), and HBDT (ESORICS 2024).
2. **Memory and Communication Analysis:** We have added a new section (Sec. 5.3) to discuss memory and communication overhead (key size, ciphertext size) of the HeLutNet model compared to current SOTA works.
3. **Hardware-Independent Computation Cost Metric:** We have reported the total number of homomorphic operations of HeLutNet in Sec. 3.4 to provide a hardware-independent measure of computational cost.
4. **Additional Benchmark Machine Setup:** We have added latency results on two new machine setups: a legacy server (GCP N1 VM) and a modern desktop (Intel Core i7-13700K) in Appendix A.4 in addition to a modern server setup (GCP C4 VM) previously used. These new setups enable latency comparison with all prior works that all use different CPU models.
5. **Additional FHE-backend Library:** We have implemented HeLutNet using the Lattigo library in addition to the original Microsoft SEAL-based implementation to allow direct comparison to recent SOTA works that use Lattigo. The new benchmark results have been added to Tab. 3 and the new Appendix A.4.
6. **HeLutNet BGV-based vs TFHE-based Implementation:** We have included a new Appendix A.5 to compare HeLutNet's BGV-SIMD approach against HeLutNet's TFHE-PBS implementation of the same LUT network in terms of latency and total number of homomorphic operations.

We believe these revisions have fully addressed all reviewers’ concerns and suggestions. We appreciate your careful consideration of the revised manuscript and look forward to your final assessment, as well as any further questions or comments you may have.

---

### Meta-Review · Area_Chair_G6qw · 2026-01-12

**Summary:**

Overall, the evaluations tilt toward rejection, with one reviewer recommending rejection, one expressing a weak reject, and two indicating weak acceptance. The paper’s main strengths are its BGV-based SIMD implementation of LUT networks, extremely low single-sample latency, and broad evaluation across many lightweight datasets, demonstrating a clear advance in real-time FHE inference for small models. However, reviewers raised substantial concerns regarding limited novelty relative to prior LUT-based FHE work, fairness of latency comparisons given drastically simpler model architectures, and the lack of convincing scalability to more complex datasets such as CIFAR-10. While the rebuttal added extensive new experiments, comparisons, and clarifications, core concerns about practical impact, model expressiveness, and positioning relative to contemporary FHE research directions remain only partially resolved.

**Reviewer Concerns:**

The authors strengthened the paper by adding missing SOTA comparisons (TT-TFHE, Homomorphic WiSARDs, HBDT), detailed communication and memory overhead analysis, explicit reporting of homomorphic operation counts, and a direct BGV-SIMD vs. TFHE-PBS ablation. Several reviewers’ methodological concerns—such as unclear latency measurement, missing memory metrics, and unfair comparisons (e.g., DSHE)—were acknowledged and corrected. The rebuttal also clarified the security model (input privacy only), justified the choice of BGV over CKKS, and expanded evaluation beyond MNIST to a diverse set of lightweight vision, speech, health, and tabular datasets. These changes clearly improved rigor, transparency, and empirical completeness.

Despite the expanded experiments, major reviewers remain unconvinced about the practical significance and novelty of the contribution. Concerns persist that the reported speedups are largely driven by extremely shallow, low-capacity models, making comparisons to deeper FHE-friendly networks inherently mismatched. The inability to demonstrate meaningful progress toward datasets like CIFAR-10 is viewed as a fundamental scalability limitation rather than a temporary omission. Some reviewers also question whether the paper is sufficiently positioned against recent CKKS/BGV functional bootstrapping and LUT literature, and whether achieving ultra-low latency on small datasets meaningfully advances the broader PPML landscape. As a result, novelty and long-term impact remain contested.

**Reviewer Scores:**

Reviewer tHCh: Likely unchanged at 4 — acknowledged improvements and new comparisons, but original scalability and novelty concerns remain.

Reviewer ikbp: Unchanged at 2 — explicitly stated he would keep the original evaluation despite the rebuttal.

Reviewer a1MN: Likely stable at 6 — key requested comparisons were added, but concerns about expressivity and scope persist.

Reviewer iTsM: Likely stable at 6 — rebuttal addressed most technical questions, but fairness and scalability concerns remain partially unresolved.

---

### Decision · Program_Chairs · 2026-01-26

Reject